# Networks of necessity: Simulating COVID-19 mitigation strategies for disabled people and their caregivers

**Thomas E. Valles**[1,2], **Hannah Shoenhard**[3], **Joseph Zinski**[3], **Sarah Trick**[4], **Mason A. Porter**[2,5], **Michael R. Lindstrom**[2]\*

**1** Department of Mathematics, University of California, San Diego, San Diego, California, United States of America, **2** Department of Mathematics, University of California, Los Angeles, Los Angeles, California, United States of America, **3** Department of Cell and Developmental Biology, University of Pennsylvania, Philadelphia, Pennsylvania, United States of America, **4** Assistant Editor at tvo.org (TVOntario), Toronto, Ontario, Canada, **5** Santa Fe Institute, Santa Fe, New Mexico, United States of America

\* mikel@math.ucla.edu

**Data Availability Statement:** Code for our simulations is available at https://bitbucket.org/3k1m/covid19-disabledcaregiverstudy/src/master/.

## Abstract

A major strategy to prevent the spread of COVID-19 is the limiting of in-person contacts. However, limiting contacts is impractical or impossible for the many disabled people who do not live in care facilities but still require caregivers to assist them with activities of daily living. We seek to determine which interventions can best prevent infections of disabled people and their caregivers. To accomplish this, we simulate COVID-19 transmission with a compartmental model that includes susceptible, exposed, asymptomatic, symptomatically ill, hospitalized, and removed/recovered individuals. The networks on which we simulate disease spread incorporate heterogeneity in the risk levels of different types of interactions, time-dependent lockdown and reopening measures, and interaction distributions for four different groups (caregivers, disabled people, essential workers, and the general population). Of these groups, we find that the probability of becoming infected is largest for caregivers and second largest for disabled people. Consistent with this finding, our analysis of network structure illustrates that caregivers have the largest modal eigenvector centrality of the four groups. We find that two interventions—contact-limiting by all groups and mask-wearing by disabled people and caregivers—most reduce the number of infections in disabled and caregiver populations. We also test which group of people spreads COVID-19 most readily by seeding infections in a subset of each group and comparing the total number of infections as the disease spreads. We find that caregivers are the most potent spreaders of COVID-19, particularly to other caregivers and to disabled people. We test where to use limited infection-blocking vaccine doses most effectively and find that (1) vaccinating caregivers better protects disabled people from infection than vaccinating the general population or essential workers and that (2) vaccinating caregivers protects disabled people from infection about as effectively as vaccinating disabled people themselves. Our results highlight the potential effectiveness of mask-wearing, contact-limiting throughout society, and strategic vaccination for limiting the exposure of disabled people and their caregivers to COVID-19.

The other data that we used are publicly available and cited in our paper.

**Funding:** MAP acknowledges support from the National Science Foundation (grant number DMS-2027438) through the RAPID program. The website for the National Science Foundation is https://www.nsf.gov. The funders had no role in study design, data collection and analysis, decision to publish, or preparation of the manuscript.

**Competing interests:** We declare the following competing interests: HS provides care, JZ and ST receive care, and ST reports on the COVID-19 pandemic as a journalist.

## Author summary

Disabled people who need help with daily life tasks, such as dressing or bathing, have frequent close contacts with caregivers. This prevents disabled people and their caregivers from physically distancing from one another, and it also significantly increases the risk of both groups of contracting COVID-19. How can society help disabled people and caregivers avoid infections? To answer this question, we simulate infections on networks that we model based on a city of about one million people. We find that one good strategy is for both disabled people and their caregivers to use masks when they are together. We also find that if only disabled people limit their contacts while other people continue their lives normally, disabled people are not protected effectively. However, it helps disabled people substantially if the general population also limits their contacts. We also study which vaccination strategies can most efficiently protect disabled people. Our simulations suggest that vaccinating caregivers against COVID-19 protects the disabled subpopulation about equally effectively as vaccinating a similar number of disabled people. Our findings highlight both behavioral measures and vaccination strategies that society can take to protect disabled people and caregivers from COVID-19.

## 1 Introduction

The coronavirus disease 2019 (COVID-19) pandemic, which is caused by the severe acute respiratory syndrome coronavirus 2 (SARS-CoV-2) virus, has revealed major societal vulnerabilities in pandemic preparation and management [1]. Existing social disparities and structural factors have led to a particularly adverse situation for the spread of COVID-19 in vulnerable groups. Therefore, it is crucial to examine how to mitigate its spread in these vulnerable groups [2] both to address these difficulties in the current pandemic and to prepare for future pandemics [3]. The effectiveness of society-wide behavioral interventions in mitigating viral spread in the general population is now well-documented [4–8]. However, the effectiveness of these non-pharmaceutical interventions (NPIs) has not been assessed in certain vulnerable groups. One such group is disabled people, who may choose to live in a group-care setting (such as a nursing home) or live independently with some caregiver support. It has been speculated that the latter arrangement increases the risk of disabled people to exposure to infections [9]. However, to the best of our knowledge, this situation has not been studied using epidemiological modeling. Vaccinations have also been extraordinarily effective at mitigating COVID-19; they have decreased case numbers and case rates, onset of symptomatic disease, hospitalizations, and mortality numbers and rates [10–16]. However, strategies for how to most efficiently use vaccines to protect independently-housed disabled people have not yet been evaluated. In the present paper, we study a compartmental model of COVID-19 spread on a network to examine the effectiveness of several non-pharmaceutical interventions (NPIs) and vaccination strategies to mitigate the spread of COVID-19 among independently-housed disabled people and their caregivers.

People with disabilities who require assistance with activities of daily living (ADLs) may live in a long-term care facility or independently with some form of caregiving support [17, 18]. Although extensive epidemiological and modeling studies have identified risk factors and mitigation strategies for COVID-19 outbreaks in long-term care facilities [19–25], there have not been similar studies of independently-housed disabled people and their caregivers. Caregivers are often indispensable for the health and independence of disabled people because they

assist with activities such as bathing, dressing, and using the bathroom. However, in a pandemic, public-health concerns dictate that it is important to minimize in-person contacts. Disabled people and their caregivers thus face an urgent question: How can they continue to interact while minimizing the risk of COVID-19 transmission?

This question is especially urgent because of the high prevalence of risk factors for severe COVID-19 in the disabled population. These risk factors, for which we give statistics for adults of ages 45–64 in the United States (see Fig 1) [26, 27], include obesity (about 46.7% of adults with a disability have a body mass index (BMI) that indicates obesity, compared with about 31.7% of adults without a disability), heart disease (15.0% of adults with a disability and 4.6% of adults without one), Chronic Obstructive Pulmonary Disease (COPD) (20.5% of adults with a disability and 3.7% of adults without one), and diabetes (25.6% of adults with a disability and 10.6% of adults without one). A recent study reported that disabled people have a roughly 60% higher risk of death if hospitalized due to COVID-19 than people who are not disabled [28]. Additionally, whatever factor or factors initially caused a person's disability can also complicate medical management of their case if they contract COVID-19. Furthermore, isolating while ill can be impossible for disabled people because they rely on caregivers to assist them with essential daily tasks. This can make disabled people more prone to spread COVID-19 to caregivers if they contract it. Consequently, preventing COVID-19 infection in disabled and caregiver populations should be a high priority.

Caregivers also experience high risk of exposure to and death from COVID-19. Caregiving workers are disproportionately likely to be women, immigrants, and people of color. The median wage for in-home caregivers is $12.12 per hour, and their median annual earnings are $17,200 (which is below the U.S. federal poverty guideline for a two-person household) [29]. Experiencing poverty or being Black or Latinx independently increase the risk because of systemic disadvantages in accessing healthcare [30–32]. Furthermore, the COVID-19 pandemic has brought immense challenges to the caregiving workforce, including frequent lack of personal protective equipment (PPE), pandemic-specific training, paid time off, and childcare

**Fig 1. The comorbidity rates that predispose individuals (of ages 45–64) to severe cases of COVID-19 among adults in the United States without (blue) and with (pink) disabilities.**

[29]. Finally, much caregiving work is impossible without close physical contact, which elevates caregivers' risk of occupational exposure. In summary, caregivers are often members of groups that are at higher risk both of COVID-19 exposure and of more severe illness from it.

According to a 2018 report [33], approximately 26% of United States adults (including about 41% of those who are 65 or older) have some form of disability. In 2016, Lauer and Houtenville [34] reported that 7.3% of the U.S. population have a cognitive or physical disability that causes difficulty in dressing, bathing, or getting around inside their home. (We acknowledge the large uncertainty in this estimate.) At least 2.4 million people in the U.S. (i.e., approximately 0.7% of the population) are employed as home-care workers, but this is likely an underestimate because of the difficulty of accurate statistical collection [29]. An intense time commitment and irregular hours are necessary for care, so many disabled people hire multiple caregivers and many caregivers work for multiple disabled people [35]. Therefore, there is significant potential for the spread of COVID-19 in and between these two vulnerable groups, making it a high priority to identify effective methods to reduce COVID-19 spread among disabled people and caregivers without compromising care.

To mitigate disease spread during a pandemic, governments may choose to implement society-wide shutdown orders, mask mandates, and/or physical-distancing guidelines. However, governments in some regions have been reluctant to issue such orders, and some people may not fully comply with them. This raises the issue of what disabled people and caregivers can do to protect themselves both with and without society-wide pandemic-mitigation efforts. With this in mind, we test how effectively mask-wearing (i.e., using PPE), limiting the caregiver pool sizes of disabled people, and limiting contacts of disabled people can prevent COVID-19 infections when the general population either maintains their normal contact levels or limits them. To the best of our knowledge, this is the first time that mathematical modeling has been used to evaluate these issues for COVID-19 infections.

Multiple COVID-19 vaccines are now widely available in some countries, but vaccine supplies remain scarce in other countries. As of late August 2021, only 1.6% of people in low-income countries had received at least one dose of any COVID-19 vaccine [36]. Furthermore, other pandemics may emerge in the future. Consequently, it is valuable to evaluate how to most effectively allocate a small number of vaccine doses to protect vulnerable groups (such as disabled people). Specifically, we investigate whether vaccinating disabled people or caregivers is more effective than other vaccination strategies at reducing the total number of cases in these two vulnerable groups.

In this paper, we simulate COVID-19 spread on model networks that represent a city. We base the parameter values in these networks on Ottawa, Canada. Our stochastic model of disease spread incorporates several disease states (i.e., "compartments"), different occupation types in a population, the heterogeneity of the risk levels across different interactions, and time-dependent lockdown measures. Our disease-spread model, which we explain in Section 2, allows us to quantitatively study our various questions under our set of assumptions. Using both calculations of structural features of our networks and simulations of the spread of a disease on our networks, we find that disabled people and caregivers are both substantially more vulnerable to COVID-19 infection than the general population (perhaps because of their large network centralities). We test the effectiveness of several NPIs—including limiting the number of social contacts, wearing masks, and limiting the number of caregivers that a given disabled person sees—at preventing COVID-19 spread in disabled and caregiver populations. By selectively seeding infections or blocking infections (via a simulated vaccine) in certain groups, we identify caregivers as major drivers of COVID-19 spread—especially for disabled people and their caregivers—and suggest that caregivers should be prioritized in vaccination campaigns.

Our paper proceeds as follows. We present our stochastic model of the spread of COVID-19 in Section 2, our results and a series of case studies in Section 3, and our conclusions and further discussion in Section 4. We describe the details of our model in the S1 Text.

## 2 A stochastic model of the spread of COVID-19 infections

We start by giving a rough idea of our stochastic model of the spread of COVID-19, and we then discuss further details in Section 2.2. Readers who are interested predominantly in the essence of our model can safely skip Section 2.2. We give a comprehensive list of our assumptions in Section 2.3. Readers who wish to use our code can find it at a Bitbucket repository. We previously wrote a white paper about this topic [37]; the present manuscript gives the full details of our study.

### 2.1 A brief overview of our model

Numerous researchers have used mathematical approaches to examine the spread of COVID-19 [38, 39]. Such efforts have used a variety of frameworks, including compartmental models [40, 41], self-exciting point processes [42, 43] (which one can also relate to some compartmental models [44]), and agent-based models [45]. Many of these models incorporate network structure to examine how social contacts affect disease spread. Some models have incorporated age stratification [46], mobility data and other data to help forecast the spread of COVID-19 [44, 47, 48], and/or the structure of travel networks [49]. In the present paper, we use an agent-based approach to study COVID-19 within a single city. Our approach involves simulating a stochastic process on time-dependent networks [50, 51]. One of the features of our model is that different segments of the population have different degree distributions, with mixing between these different segments. To examine networks with these features, we use generalizations of configuration models [52, 53].

In our model population, we consider three types of interactions between individuals, six disease states, and four distinct groups (i.e., subpopulations). We encode interactions using a network, and all interactions between different individuals involve exactly two people. We suppose that *strong* interactions describe interactions at home within family units (or, more generally, within "household units"); that *weak* interactions describe social interactions and interactions that take place at work, at a grocery store, and so on; and that *caregiving* interactions describe interactions between caregivers and the disabled people for whom they care. We model each of these interaction types with a different baseline risk level of disease transmission. Weak interactions have the lowest baseline risk level, strong interactions have the next-lowest baseline risk level, and caregiving interactions have the highest baseline risk level.

We use a compartmental model of disease dynamics [54], which we study on contact networks [55, 56]. We assume that our population (e.g., of a single city, like Ottawa) is closed and that each individual is in exactly one disease state (i.e., compartment). Our model includes *susceptible (S)* individuals, who can contract COVID-19; *exposed (E)* individuals, who have the disease but are not yet infectious or symptomatic; *asymptomatic (A)* individuals, who do not have symptoms but can spread the disease; *ill (I)* individuals, who are symptomatically ill and contagious; *hospitalized (H)* individuals, who are currently in a hospital; and *removed (R)* individuals, who are either no longer infectious or have died from the disease. The A compartment includes prodromal infections, asymptomatic individuals, and mildly symptomatic individuals; in all of these situations, an individual has been infected, but we assume that they are not aware of it. Our model does not incorporate loss of immunity or births, and we classify both "recovered" and removed individuals as part of the R compartment. In our study, an

individual has been infected if they are no longer in the S compartment. Therefore, cumulative infections include every individual that is currently in the E, A, I, H, or R compartments.

We divide our model city's population into the following subpopulations:

- *caregivers*, who provide care to disabled people;

- *disabled people*, who receive care;

- *essential workers*, whose occupations prevent them from limiting contacts during lockdowns and similar policies, but who are not already included in the caregiver subpopulation; and

- the *general population*, which is everyone else.

The individuals in the disabled subpopulation have two types of caregivers: *weak* caregivers, who are professional caregivers, for whom a caregiving contact with a disabled person is likely to break if either individual in the relationship becomes symptomatic; and *strong* caregivers, for whom a caregiving relationship persists even if the individuals in it are symptomatic (as long as neither individual is hospitalized). We consider these two types of caregivers to account for family members or close friends who always provide some care to a disabled person. Although our model includes a hospitalized compartment, we do not model doctors, nurses, custodial services, or other hospital staff who are involved in caring for COVID-19 patients. The caregivers in our model population refer strictly to individuals who provide supportive assistance to members of the disabled community in their homes. We also do not model skilled care facilities, such as nursing homes.

When an individual is symptomatic, we assume that they distance themselves from society (by breaking contacts) with a fixed probability $b \in [0, 1]$. The probability can be less than 1 to account for a variety of situations, such as people who feel financial pressure to work anyway [57], people who have symptoms that are so mild that they are unaware of them, and people who ignore common decency. In our model, the breaking of contacts of an individual who becomes ill means that they temporarily cut off all weak contacts and (if relevant) weak caregiver–disabled relationships and only maintain contacts within their household unit and (if relevant) their strong caregiver–disabled relationships until they recover. If an individual becomes hospitalized, these strong contacts also break.

We seek to understand how COVID-19 spreads with time in caregiver, disabled, essential-worker, and general populations and how different mitigation strategies, such as contact-limiting and mask-wearing, affect the outcome of disease spread. Consequently, we allow the distributions of the numbers of contacts to change with time and adjust the disease transmission probability to reflect the presence of masks.

We tune our baseline model to describe the city of Ottawa from its first reported case on 10 February 2020 [58] through its closure of non-essential businesses on 24 March 2020 [59] (the closure order occurred on 23 March) and then to understand how its "Phase 1" reopening on 6 July 2020 [60] affected disease spread. In Fig 2, we illustrate an egocentric network (i.e., "ego network") [61] that is centered at a single disabled person in the population before and after closure.

## 2.2 Specific details of our model

We now give a detailed description of our model. One of the key features of the networks on which a disease spreads is the numbers and distributions of the contacts of different types of individuals. We incorporate these features by constructing networks using a generalization of configuration-model networks. See [62] for a review of configuration models.

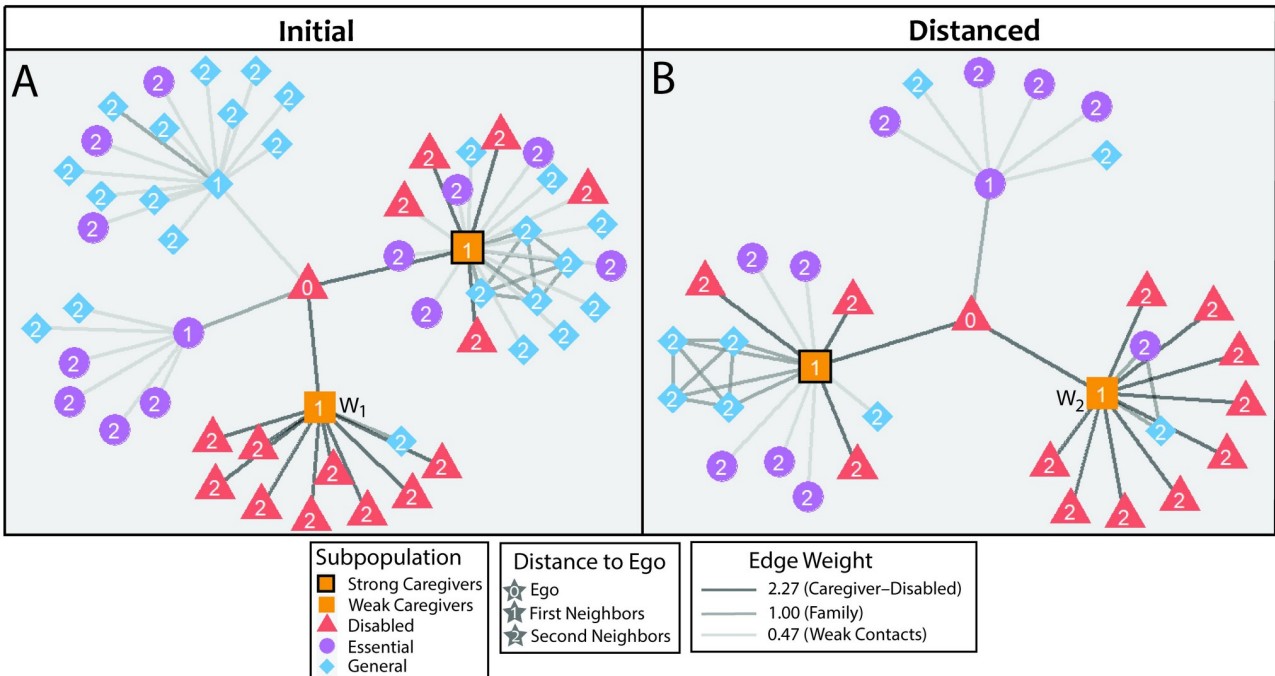

**Fig 2. An egocentric network (i.e., ego network) of an example disabled person on (A) day 43 (before the start of contact-limiting) and (B) day 45 (during contact-limiting).** The two ego networks encode contacts for the same disabled person. The label 'W$_1$' denotes the weak caregiver on day 43 and the label 'W$_2$' denotes the weak caregiver on day 45. In this example, W$_1$ and W$_2$ are different caregivers. We illustrate the different groups (colors) in our model city, the interaction strengths between individuals (line thicknesses), and distances (numbers) from the ego. The edge weights are relative to the strong-contact weight of 1.

To each node (i.e., individual) in a network, we assign a group (disabled, caregiver, essential worker, or general) and then assign both weak contacts and strong contacts. Additionally, we assign caregiver nodes to each disabled node and assign disabled nodes to each caregiver node. No individual is both a strong contact and a weak contact of the same person, no individual is both a caregiver for a disabled person and an ordinary weak contact of that disabled person, and so on. We anticipate a large variance in the number of weak contacts, with some people having many more contacts than others [63], so we assign each individual a number of weak contacts from an approximate truncated power-law distribution (see Section B of the S1 Text). Because strong contacts represent household units, we assign each individual a number of strong contacts from an empirical distribution that we construct using census data of household sizes in Canada [64]. To model the pools of caregivers that are available to disabled people, we assume that each disabled node has a fixed number of weak caregivers (this pool does not change with time) and that this fixed number is the same for all disabled nodes. We were unable to find reliable data about the sizes of caregiver pools, so we make an educated guess that is consistent with the lived experience of the disabled authors of the present paper. We also assign one strong caregiver to each disabled node.

The contact structure in our networks can change with time. For example, weak contacts may break if a lockdown starts, both weak and strong contacts break when an individual is hospitalized, and so on. Each day, we choose one member of a disabled individual's caregiver pool uniformly at random to potentially provide care to them. (It is only potential care because that caregiver may have broken contact due to illness.) Each day, the disabled individual also receives care from a single strong caregiver, if possible. (This occurs as long as that contact has not been broken due to hospitalization.) In each time step, which consists of one day, the

disease state (i.e., compartment) of an individual can change. From one day to the next, we compute the transition probability from susceptible to exposed using Eqs (1) and (2) and each susceptible individual's disease state at the start of the day. On each day, we determine transitions between different disease states by generating exponential random variables for transition times. When a generated transition time occurs within a 1-day window, an individual changes compartments. If two different transitions are possible and both exponential random variables are less than 1 day, we use the state transition that corresponds to the shorter transition time. Individuals who break their contacts because of illness do so immediately upon entering a new compartment. Any network restructuring occurs at the start of a day (i.e., before we calculate exposure risks).

In the S1 Text, we give the day-to-day time evolution of our model in Algorithms 1, 7, and 8 and the network-construction process in Algorithms 3, 4, 5, and 6. We host our code at a Bitbucket repository.

When we construct our networks, we use three families of discrete probability distributions:

- The distribution $\mathcal{P}(a_-, a_+, \mu)$ is an approximate truncated power-law distribution. If $N \sim \mathcal{P}(a_-, a_+, \mu)$, then $N$ takes integer values in $\{a_-, a_- + 1, \ldots, a_+\}$. For large $n$, we have that $\Pr(N = n) = O(1/n^p)$, where we choose $p$ so that $\mathbb{E}(N) = \mu$. See the S1 Text for full details.

- The distribution $\mathcal{E}(p_0, p_1, p_2, \ldots, p_k)$ is a discrete distribution. If $N \sim \mathcal{E}(p_0, p_1, p_2, \ldots, p_k)$, then $\Pr(N = n) = p_n$ when $n \in \{0, \ldots, k\}$ and $\Pr(N = n) = 0$ otherwise.

- The distribution $\mathcal{F}(k)$ is a deterministic distribution that has only one attainable value. If $N \sim \mathcal{F}(k)$, then $\Pr(N = n) = \delta_{n,k}$, where $\delta$ denotes the Kronecker delta (which equals 1 if the subscripts are equal and 0 if they are not).

Our model has three key dates: the *first recorded case*, which we set to be day 1 (i.e., 10 February 2020), as we use day 0 for initial conditions to produce a seed case (which is in the asymptomatic compartment) of the disease; *lockdown* (i.e., 24 March 2020), which is when contact-limiting begins and some individuals start wearing masks; and *reopening* (i.e., 6 July 2020), which is when the city begins to reopen. In the S1 Text, see Algorithm 9 for how we implement a lockdown and Algorithm 10 for how we implement reopening. For mask-wearing, we focus on four situations:

- None (N): nobody wears a mask.

- Disabled people and caregivers wear masks (D+C): disabled people and caregivers both wear masks when interacting with each other, but nobody else wears a mask.

- Disabled people, caregivers, and essential workers wear masks (D+C+E): all of the mask-wearing in the (D+C) scenario occurs, and we also assume that both individuals in an interaction wear a mask whenever there is a weak interaction with an essential worker (to model interactions in places like grocery stores, banks, and doctors' offices during routine visits).

- All weak contacts wear masks (All*): the same individuals under the same conditions as in (D+C+E) wear masks, and we also assume that both individuals wear a mask in any interaction between weak contacts.

To model essential workers, we assume (except when there are symptoms of illness) that weak contacts with essential workers are not broken. Therefore, during a lockdown, essential workers continue to have a large number of weak contacts on average. We similarly

characterize the caregiver subpopulation; they retain their interactions with their associated disabled nodes. We assume that an individual's weak contacts during a lockdown are a subset of their weak contacts from before a lockdown. Upon a reopening, each individual is assigned a new number of weak contacts. They retain the weak contacts that they had during a lockdown, but they can also gain new weak contacts that they did not possess before the lockdown if their new number of weak contacts is larger than their number of weak contacts immediately prior to reopening. For example, if an individual has 3 weak contacts during a lockdown and is assigned 7 weak contacts after reopening, then they need 4 weak contacts. These 4 new weak contacts can be different from that individual's weak contacts before the lockdown. We do this to account for situations (such as business closures or job loss) that cause individuals to visit different stores or workplaces after a city reopens.

We discretize time into units of $\Delta T = 1$ day. Our model, which one can view as an agent-based model, evolves as the individuals interact with other. Individuals who are in the S compartment can move into the E compartment, depending on their interactions on a given day. Each day, the probability that susceptible individual $i$ remains in the S compartment is

$$\sigma_i = \prod_{j \in B(i)} \left(1 - \beta w_w^{W_{ij}} w_c^{C_{ij}} m^{M_{ij}/2}\right), \tag{1}$$

where $\beta$ is the baseline transmission probability, $w_w$ is the edge weight of a weak contact, $w_c$ is the weight of a caregiving contact, $m$ is the risk reduction from mask-wearing by both individuals in an interaction, $B(i)$ is the set of all active (i.e., non-broken) contacts of individual $i$ that are infectious, $W_{ij}$ is 1 if $i$ and $j$ are weak contacts and 0 otherwise, $C_{ij}$ is 1 if $i$ and $j$ have a caregiving relationship and 0 otherwise, and $M_{ij}$ (which can be equal to 0, 1, or 2) counts how many of individuals $i$ and $j$ wear a mask during an interaction between them. The term $\beta w_w^{W_{ij}} w_c^{C_{ij}} m^{M_{ij}/2}$ gives the probability that node $i$ becomes infected from an interaction with node $j$. Given $\sigma_i$, we compute the probability that $i$ transitions from the susceptible compartment (S) to the exposed compartment (E) in a given day:

$$\Pr(i \text{ transitions from S to E}) = 1 - \sigma_i. \tag{2}$$

We model the outcomes of transitions from E to A, transitions from A to I, transitions from A to R, transitions from I to H, transitions from I to R, and transitions from H to R as exponential processes with fixed rates of $\nu$, $\alpha$, $\eta$, $\mu$, $\rho$, and $\zeta$, respectively. In our simulations, transitions occur in intervals of size $\Delta T$ (which we set equal to 1 day, as mentioned previously). When multiple transitions between compartments are possible, such as from A to I and from A to R, we treat event transitions as competing exponential processes. (See Algorithms 1, 7, and 8 of the S1 Text.) We summarize the possible state transitions and their rates in Fig 3.

In our simulations, we uniformly-at-random initialize a fixed number $A_0$ individuals to be asymptomatic on day 0. To account for limited testing availability in the early stages of the pandemic, we assume that only a fraction $\tau$ the individuals in the I compartment (i.e., they are symptomatically ill but not hospitalized) have a positive COVID-19 test. Having a positive COVID-19 test means that an individual has a *documented* case of COVID-19. At initialization, we determine whether or not an individual will test positive if they are symptomatically ill by assigning them a true/false flag $P$ with probability $\tau$ for true. When $P$ is true, if that individual becomes symptomatic (i.e., enters the I compartment), we suppose for simplicity that they have a positive COVID-19 test immediately upon entering the I compartment (i.e., before the next day begins). When $P$ is false, that individual only has a positive COVID-19 test if they are hospitalized. For simplicity, we assume that their positive test takes place immediately upon entering the H compartment. We also assume that we do not double-count individuals

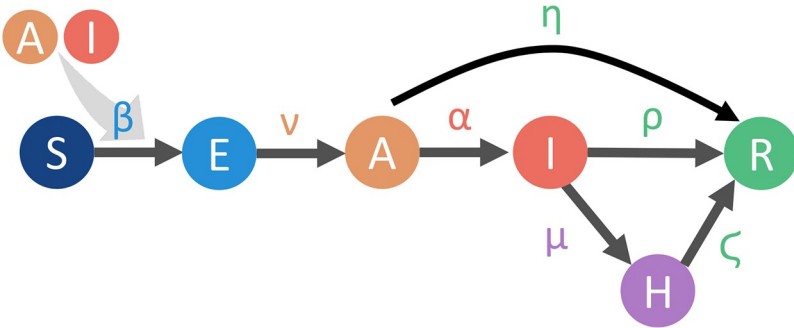

**Fig 3. A schematic illustration of our compartmental model of disease transmission.** Susceptible individuals (S), by being exposed to asymptomatic (A) or symptomatically ill (I) individuals, can become exposed (E) with a baseline transmission probability $\beta$. One can reduce the risk level of an interaction through the NPI of mask-wearing; this multiplies the risk level by the factor $m^{1/2}$ (if only one individual in the interaction wears a mask) or the factor $m$ (if both individuals in the interaction wear a mask). Caregiving interactions have a higher risk level (by a factor $w_c$) than the baseline and weak interactions have a lower risk level (by a factor $w_w$) than the baseline. Exposed individuals are not yet contagious; however, they can eventually transition to the asymptomatic state. From the asymptomatic state, an individual can either become symptomatically ill or be removed (R), which encompasses recovery, death, and any other situation in which an individual is no longer infectious. If an individual is symptomatic, they can either be removed or become hospitalized (H). From the hospitalized state, an individual eventually transitions to the removed state. The state-transition parameters that we have not yet mentioned are fixed rates of exponential processes.

who have a positive COVID-19 test while in the I compartment if they later enter the H compartment. We suppose that no asymptomatic infections are documented in the early spread of the disease. We compute daily tallies of cumulative documented cases at the end of each day. We need the assumption of not having positive COVID-19 tests of all infected individuals to be able to fit our parameters to the Ottawa data, which gives the number of documented cases (but not the cumulative number of total infections) as a function of time.

In Table 1, we present the parameters that we use in our model. We discuss and support the values of these parameters in Section A of the S1 Text. Whenever possible, we seek to infer parameters directly from clinical data, instead of simply adopting our parameter values from those of existing models in the literature. The only parameter that we borrow in this way is $\nu$, which is the transition rate from the E compartment to the A compartment.

### 2.3 Summary of our assumptions

We now briefly summarize the main assumptions of our model.

**Population.**   Our model city's population is closed, so the city has no inflow or outflow.

**Time units.**   We discretize time in units of $\Delta T = 1$ day.

**Composition.**   The population of our city consists of the following types of individuals: 7.3% are disabled, 2.1% are caregivers, 14.72% are essential workers, and 75.88% are members of the general population. The "roles" (i.e., subpopulation memberships) of individuals do not change.

**Disease compartments.**   Individuals can be susceptible, exposed, asymptomatic, symptomatically ill, hospitalized, or removed. All infected individuals must go through the exposed compartment before becoming infectious.

**Transitions between compartments.**   We model an individual's daily infection rate through a probability of infection per interaction with a contagious individual, with interaction probabilities scaled up or down based on the types of interactions and the presence/absence of masks. All other transitions between compartments come from exponential processes that we consider one day at a time.

**Table 1. The parameter values that we use in our study.** In the "Source" column, *literature* indicates that we found a value directly from data in the literature, *inferred* indicates that we inferred a value based on published data in the literature, *by definition* signifies a value that we set in our model formulation, *chosen* indicates that a value is unknown but we made a choice in our model, *borrowed* indicates that we adopted a value directly from a model in the literature, and *fit* indicates that we use Ottawa case data along with other (i.e., not fit) parameters in this table to estimate a value.

| Symbol | Meaning | Value | Reference | Source |
|---|---|---|---|---|
| $f_{dis}$ | fraction of population who are disabled | 0.073 | [34] | literature |
| $f_{care}$ | fraction of population who are caregivers | 0.021 | [65] | literature |
| $f_{ess}$ | fraction of population who are essential workers | 0.1472 | [65–67] | literature |
| $f_{gen}$ | fraction of population who are part of the general population | 0.7588 | [34, 65–67] | inferred |
| $m$ | risk-reduction factor from mask-wearing by both individuals in an interaction | 0.34 | [7] | literature |
| $b$ | probability that an ill individual breaks their weak contacts | 0.92 | [68] | inferred |
| $w_w$ | weak edge weight | 0.473 | [7, 69] | inferred |
| $w_s$ | strong edge weight | 1 | N/A | by definition |
| $w_c$ | caregiving edge weight | 2.27 | [69, 70] | inferred |
| $\beta$ | baseline transmission probability | 0.0112 | [69] | inferred |
| $\nu$ | transition rate from E to A | 1 day$^{-1}$ | [71] | borrowed |
| $\alpha$ | transition rate from A to I | 0.0769 day$^{-1}$ | [72–75] | inferred |
| $\eta$ | transition rate from A to R | 0.0186 day$^{-1}$ | [72–75] | inferred |
| $\mu$ | transition rate from I to H | 0.0163 day$^{-1}$ | [73, 76–78] | inferred |
| $\rho$ | transition rate from I to R | 0.0652 day$^{-1}$ | [73, 76–78] | inferred |
| $\zeta$ | transition rate from H to R | 0.0781 day$^{-1}$ | [79] | inferred |
| $\tau$ | probability of tested if ill but not hospitalized | 0.04 | [58] | fit |
| $C^*$ | maximum number of contacts in approximate power law | 60 | [58] | fit |
| $\mathcal{D}_{strong}$ | distribution of strong contacts | $\mathcal{E}(0.283, 0.332, 0.155, 0.148, 0.0816)$ | [64] | literature |
| $\mathcal{D}_{pool}$ | distribution of pool sizes | $\mathcal{F}(10)$ | n/a | chosen |
| $\mathcal{D}_{ess,pre}$ | essential worker weak-contact distribution when not distancing | $\mathcal{P}(0, C^*, 21.37)$ | [80] | inferred |
| $\mathcal{D}_{ess,post}$ | essential worker weak-contact distribution during distancing | $\mathcal{P}(0, C^*, 21.37)$ | [80] | inferred |
| $\mathcal{D}_{gen/dis,pre}$ | general/disabled weak-contact distribution when not distancing | $\mathcal{P}(0, C^*, 10.34)$ | [80] | inferred |
| $\mathcal{D}_{gen/dis,post}$ | general/disabled weak-contact distribution during distancing | $\mathcal{P}(0, C^*, 7.08)$ | [80] | inferred |
| $\mathcal{D}_{care,pre}$ | caregiver weak-contact distribution when not distancing | $\mathcal{P}(0, C^*, 5.14)$ | [80] | inferred |
| $\mathcal{D}_{care,post}$ | caregiver weak-contact distribution during distancing | $\mathcal{P}(0, C^*, 4)$ | [80] | inferred |
| $A_0$ | number of asymptomatic individuals on day 0 | 341 | [58] | fit |
| $P_{Ottawa}$ | population of Ottawa | 994837 | [81] | literature |

**Strong contacts.** We assign the numbers of strong contacts of all individuals from the empirical probability distribution $\mathcal{D}_{strong}$.

**Weak contacts.** Given an individual's subpopulation membership and the status of contact-limiting, we determine the individual's number of weak contacts from an approximate truncated power-law distribution (see Section B of the S1 Text) using $\mathcal{D}_{group,status}$, where *group* is one of "gen/dis" (i.e., the general and disabled subpopulations), "care" (i.e., caregivers), or "ess" (i.e., essential workers) and *status* is one of "pre" (i.e., not during a lockdown) or "post" (i.e., during a lockdown). The weak-contact distribution has the same parameter values for disabled people and individuals in the general population.

**Caregiving contacts.** All disabled people have a pool of weak-contact caregivers of a size that is dictated by $\mathcal{D}_{pool}$. For each disabled person, we choose that pool uniformly at random from the set of caregivers. Additionally, each disabled person has one strong caregiver that we choose uniformly at random from the set of caregivers. They see that individual each day, unless either the disabled person or that caregiver is hospitalized.

**Breaking contacts.** Asymptomatic individuals do not break contacts (except in the form of contact-limiting). An ill (but not hospitalized) individual breaks their weak contacts with probability $b$. If an individual is hospitalized, they break both their weak contacts and their strong contacts until they move into the R compartment. An individual in the R compartment does not infect others with the disease; they may be deceased or simply no longer infectious. In our computations, those individuals regain their weak and strong contacts.

**Interactions.** Each day, an individual interacts with the same weak (except for caregiver–disabled interactions) and strong contacts unless the contact has been broken due to illness or when the contact distributions change. Each day, a disabled person interacts with their strong caregiver, unless illness prevents it. Additionally, on each day, a disabled person interacts with a uniformly randomly selected member of their caregiver pool, unless illness prevents it. Even during contact-limiting stages, the weak contacts of essential workers do not break.

## 3 Results

We first compare the new daily documented cases and the cumulative number of documented cases in our model with empirical case data from Ottawa (see Fig 4). We fit the parameters in our model up to 10 May 2020 (i.e., day 90) of the epidemic in Ottawa, and we assume that the city immediately enters a contact-limited phase on 24 March (i.e., day 44). We do the fitting (see Section A of the S1 Text) by minimizing the $\ell_2$-error in the model's count of daily documented cases. We show the 7-day mean of new daily documented cases; for each day, we

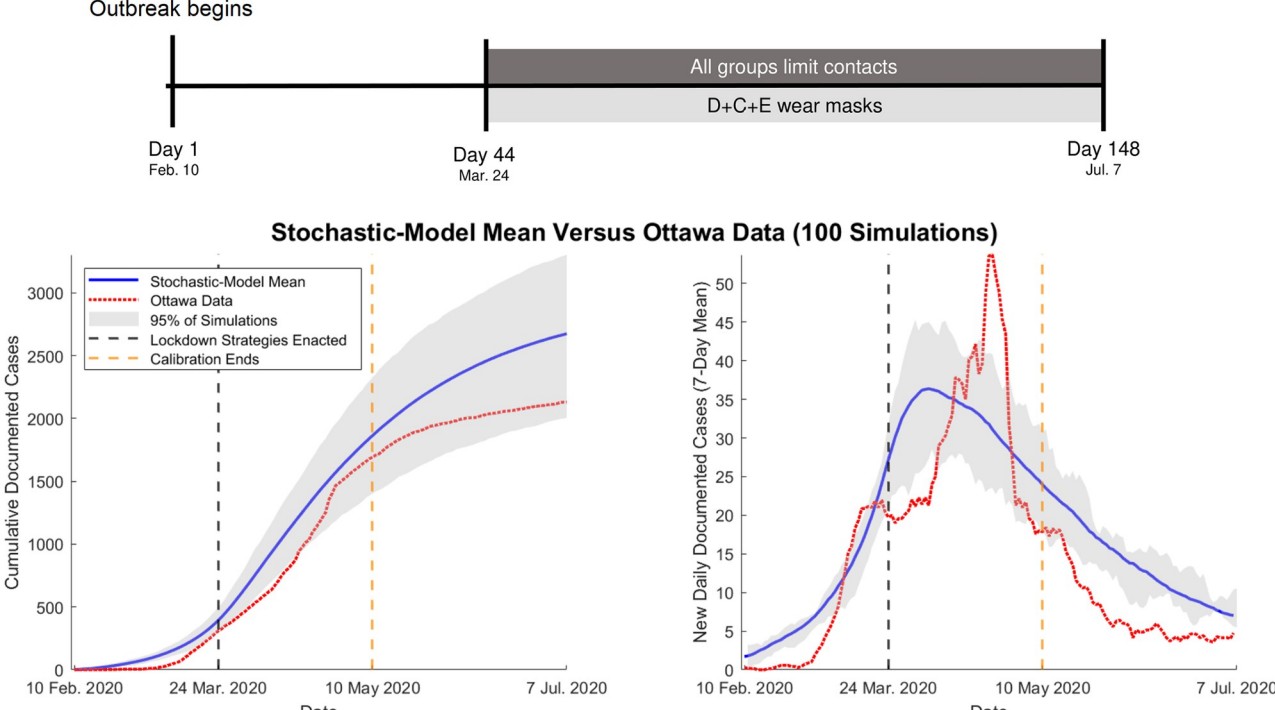

**Fig 4. A comparison of a mean of 100 simulations of our stochastic model of COVID-19 spread with (left) cumulative documented case counts and (right) the 7-day mean of new daily documented cases.** For each day, we calculate the 7-day mean over a sliding window that includes the previous three days, the current day, and the next three days. We fit the parameters by minimizing the $\ell_2$-error of the model's count of daily documented cases over the first 90 days. We show the mean of our model in blue and the Ottawa case data in red. The gray window indicates the middle 95% of these 100 simulations. On day 44 (i.e., 24 March 2020), all subpopulations limit contacts and the (D+C+E) mask-wearing scenario begins. The graphs terminate on day 148, when Ottawa had its first reopening.

calculate this mean over a sliding window that including the previous three days, the current day, and the next three days. At the endpoints, we truncate the window and take the mean over days that fill the window. We find reasonable agreement between the daily documented case counts in our model and the reported documented cases, but our match is not perfect. For example, the peak in daily documented case counts and the inflection point in cumulative documented cases occurs earlier in our model than it does in documented case records. This can arise from many possible factors, including delays in reporting cases (e.g., with differences on weekdays versus weekends), delays in the diagnosis of symptomatic individuals, changes of the model parameters (like testing availability) with time, or our use of only two degrees of freedom in our fits (with most model parameters arising from sources that are not specific to Ottawa). The daily and cumulative documented case counts in our model deviate little from the data for the first 90 days, but our model subsequently tends to overestimate the case count. We speculate that this may stem from overestimating the number of contacts of the Ottawans. Our contact estimates come from survey data [80], which do not focus specifically on Ottawa. We wish to avoid overfitting, so we accept the fit performance.

The epidemic trajectories have large variances; specifically, the 95% confidence windows are large. We believe that one of the main factors behind these large variances is our use of an approximate truncated power-law distribution for weak contacts. If we replace these approximate truncated power-law distributions with deterministic distributions (i.e., distributions with 0 variance) with the same mean values, we obtain much smaller variances and the disease also hardly spreads. We have chosen to use approximate truncated power-law distributions to allow large variations in the numbers of contacts, but this results in large variances in the epidemic trajectories. See Section C of the S1 Text for further discussion.

Our baseline transmission probability $\beta = 0.0112$ is smaller than those that were employed in some other studies [46, 82], which used $\beta \approx 0.06$. For the assumptions in our study, $\beta = 0.0112$ is appropriate. With $\beta = 0.06$, the disease is too infectious, and our simulations then result in a total documented case count that greatly exceeds the number of documented cases in Ottawa. The value $\beta = 0.06$ is also inconsistent with secondary attack-rate studies [69] when they are combined with the durations that individuals spend in the compartments of our model. The fact that the disease can still spread so readily with $\beta = 0.0112$ perhaps stems from our network structure, as some individuals can be superspreaders.

To help us understand the results of simulating our stochastic model of COVID-19 spread, we examine the structural characteristics of the networks on which we perform our simulations. Because different types of contacts have different levels of disease transmission, we base our measures on structural features of weighted networks. Additionally, our network contact structure changes with time. We compare the structure of one network from our network model on two days; one day is before contact-limiting and the other is during contact-limiting. On each of the two days, for all of the nodes in the network, we compute the numbers of first-degree contacts (i.e., direct contacts), the numbers of second-degree contacts (i.e., contacts of direct contacts), the node strengths (i.e., the sums of the edge weights, which we interpret as "conductances" of a disease across contacts), and the eigenvector centralities (i.e., the entries of the leading eigenvector of the network's adjacency matrix, where larger values of eigenvector centrality are associated with "high-traffic" nodes, which are visited often by a random walker on the network [61, 63]). We are interested in eigenvector centrality because the largest eigenvector-centrality value in a network plays a role in determining that network's susceptibility to a widespread outbreak of a disease under certain conditions [83]. We examine the distributions of the eigenvector centralities for different subpopulations in our model city. We find that caregivers and essential workers tend to have the largest numbers of first-degree and second-degree contacts, contact strengths, and eigenvector centralities (see Fig 5). The

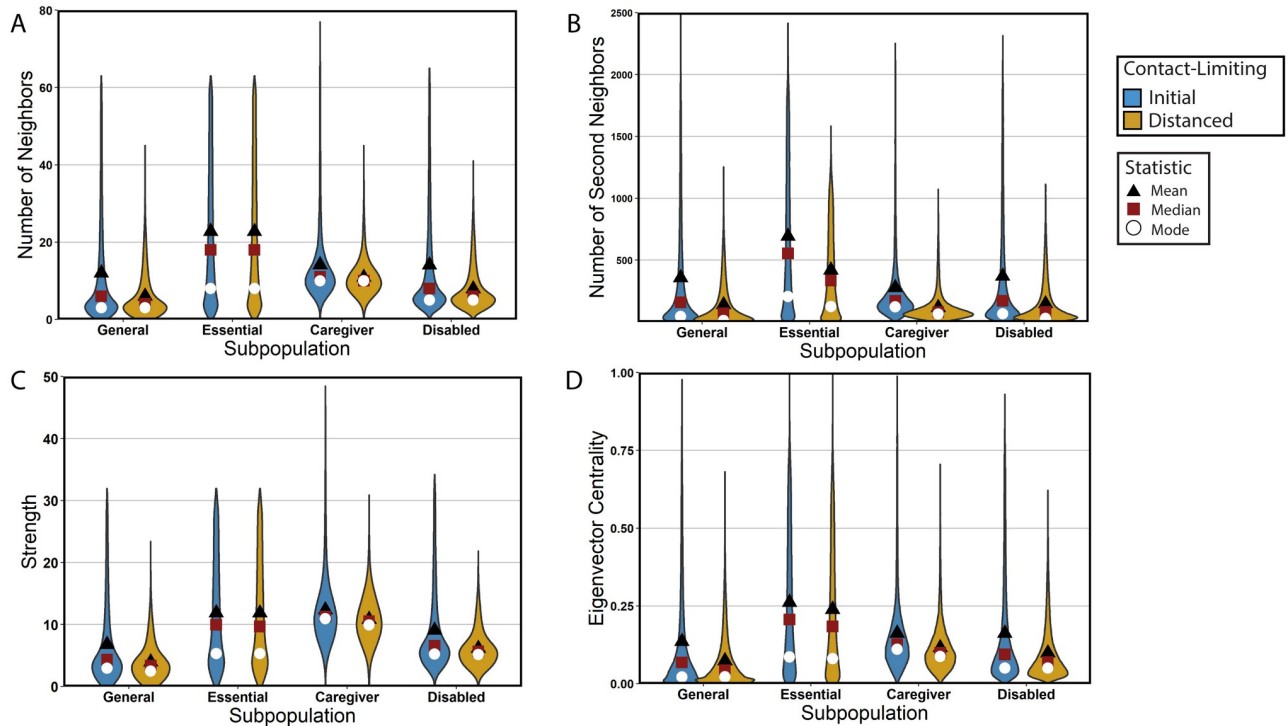

**Fig 5. Characterization of centrality measures of subpopulations in the networks on which we run our stochastic model of COVID-19 spread.** The violin plots depict empirical probability densities. The initial situation, for which we show day 43 of one simulation, has no contact-limiting. The distanced situation, for which we show day 45 of the same simulation, has contact-limiting in all subpopulations. For each subpopulation, we calculate the distributions of (A) the number of neighbors (i.e., direct contacts), (B) the number of second neighbors (i.e., contacts of contacts), (C) the strength (i.e., total edge weight) of the contacts with neighbors, and (D) eigenvector centrality.

essential-worker subpopulation has the largest mean eigenvector centrality (because of the heavy-tailed distribution of their contacts), whereas the caregiver subpopulation has the largest modal eigenvector centrality. We also test the effects of contact-limiting and mask-wearing (i.e., PPE status) strategies on the strengths (i.e., total edge weights) and eigenvector centralities in the various subpopulations. Both NPIs reduce node strengths, and contact-limiting in particular diminishes the heavy tails of the strength distributions of caregivers, disabled people, and members of the general population (see Fig 6). In other words, contact-limiting reduces the probability that individuals have a large number of contacts.

We also test how much different contact-limiting and mask-wearing strategies affect the different subpopulations in our model. We consider different mitigation strategies, which we assume are deployed on day 44, and we compare the number of cumulative infections on day 148 for these strategies. We consider the mask-wearing strategies that we outlined in Section 2.2 and the following three contact-limiting strategies:

- No contact-limiting: all people maintain their contacts for the entirety of the 148 days.

- Only disabled people limit their contacts: disabled people reduce their number of weak contacts on day 44, and all other subpopulations maintain their contacts.

- Everyone except for essential workers limits contacts: all subpopulations other than essential workers reduce their number of weak contacts on day 44.

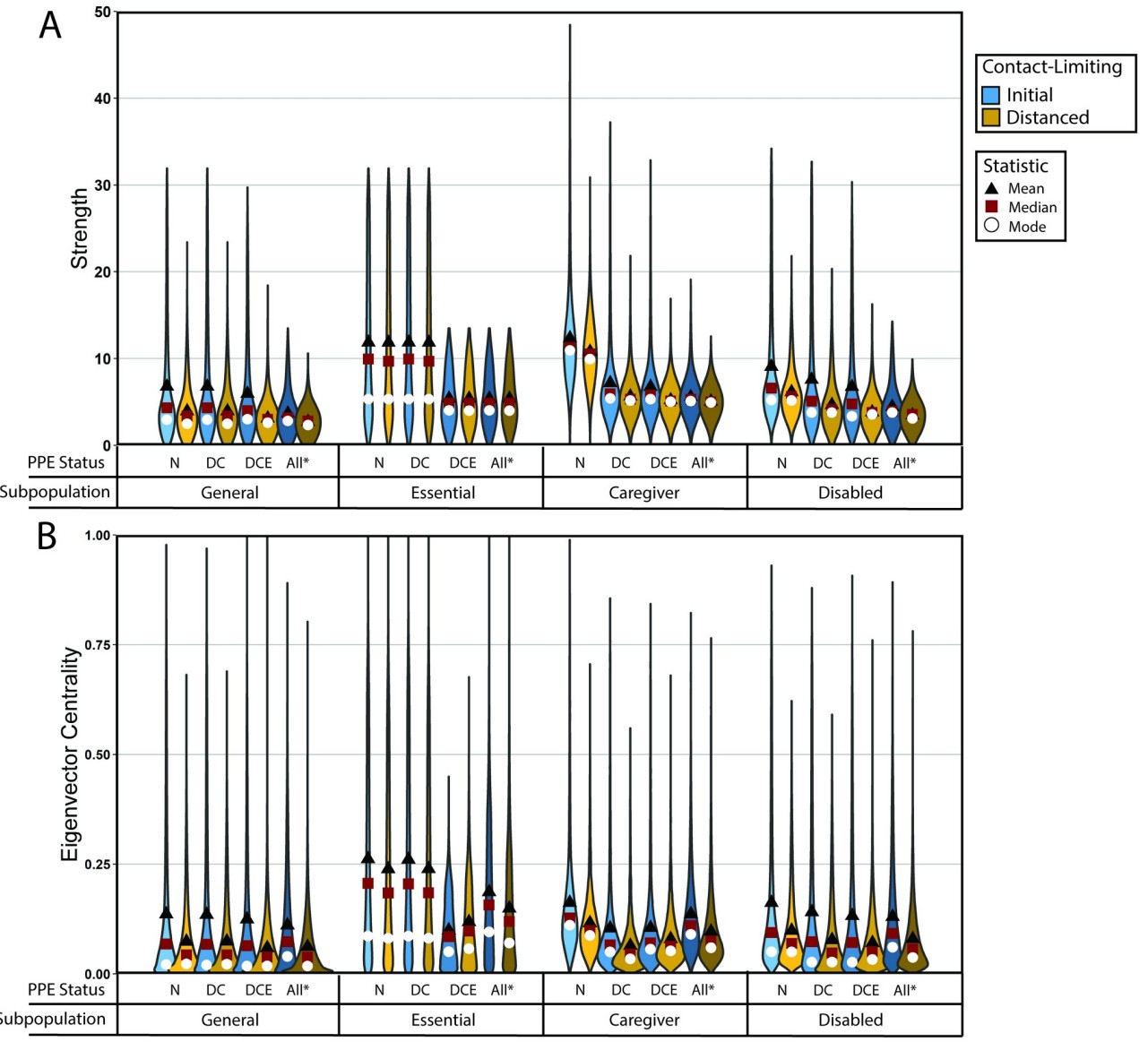

**Fig 6. Characterization of the effects of mask-wearing on centrality measures of subpopulations in the networks on which we run our stochastic model of COVID-19 spread.** The violin plots depict empirical probability densities. The initial situation, for which we show day 43 of one simulation, has no contact-limiting. The distanced situation, for which we show day 45 of the same simulation, has contact-limiting in all subpopulations. We modify edge weights by supposing that masks have the effectiveness that we indicated in Table 1. To indicate the mask-wearing statuses of different scenarios, we use the notation that we defined in Section 2.2. For each subpopulation, we compute (A) the strength distribution and the (B) eigenvector-centrality distribution.

We first consider the optimistic scenario in which all weak interactions involve mask-wearing. In this case, when everyone limits contacts on day 44, our simulations yield a mean of 13,242 cumulative infections through day 148. This is approximately 11.2% lower than the 14,910 cumulative infections through day 148 when only caregivers, disabled people, essential workers, and individuals in weak interactions with essential workers wear masks. We conclude that universal mask-wearing (specifically, in all situations except within households) is an effective NPI for reducing the number of COVID-19 cases. For all of our subsequent

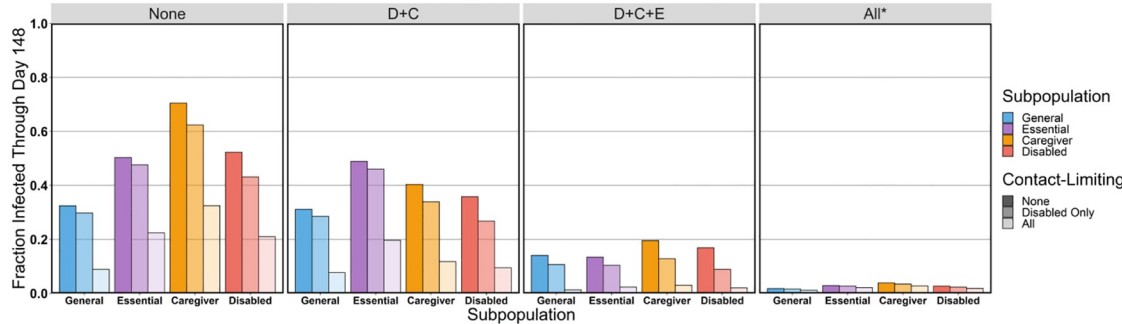

**Fig 7. The mean number of cumulative infections in the general population (blue), essential workers (purple), caregivers (orange), and disabled people (pink) for different contact-limiting and mask-wearing statuses.** The mask-wearing statuses are the same as in Fig 6.

simulations, unless we note otherwise, we assume that individuals wear masks in weak interactions only when those interactions involve essential workers.

We find that contact-limiting by only the disabled subpopulation has a relatively small effect on the number of their cumulative infections; it reduces the percent of them who become infected from 52.3% to 43.1%. Contact-limiting by only disabled people yields a similar result for caregivers, with a reduction in the percent of infected caregivers from 70.5% to 62.4%. Contact-limiting by all subpopulations has a larger effect; it reduces the percent of infected individuals in the disabled subpopulation to 21.0% and that of caregivers to 32.5%. Mask usage in both the disabled and the caregiver subpopulations protects both subpopulations even in the absence of any contact-limiting. The percent of disabled people who become infected decreases from 52.3% to 35.8%, and the percent of caregivers who become infected decreases from 70.5% to 40.3%. When essential workers, caregivers, and disabled people all wear masks, this protection is enhanced. The percent of disabled people who become infected decreases to 16.9%, and the percent of caregivers who become infected decreases to 19.5%. Finally, when all weak contacts wear masks, 2.7% of disabled people and 3.8% of the caregivers become infected. When all subpopulations limit contacts and wear masks (except within a household), 1.8% of the disabled subpopulation and 2.7% of the caregiver subpopulation become infected. We summarize the results of the mask-wearing interventions in Fig 7.

Because COVID-19 guidelines recommend reducing the number of contacts between individuals, we test whether or not reducing the number of weak caregiver contacts per pool (while maintaining daily caregiving interactions) helps protect disabled people and/or caregivers. This NPI affects the total number of contacts of disabled people, but it does not reduce the total amount of time that they are exposed to these contacts. We test caregiver pool sizes of 4, 10, and 25, and we find that reducing caregiver pool size does not reduce infections either among caregivers or among disabled people (see Fig 8).

In our investigation, we are particularly uncertain about the values of three parameters: the probability that individuals break weak contacts when they become ill, the effectiveness of masks, and the fraction of caregivers in the population. Therefore, we repeat our simulations with otherwise baseline conditions (see Table 1) for different values of these parameters. We choose the values of $m$ as the boundaries of the 95% confidence window in mask effectiveness in [7]. We choose the values of $b$ using educated guesses of reasonable best-case and worst-case scenarios. We choose the upper bound of the fraction of caregivers in the population so that caregivers have approximately the same mean number of occupational contacts as individuals in the general and disabled subpopulations. We choose the lower bound so that the

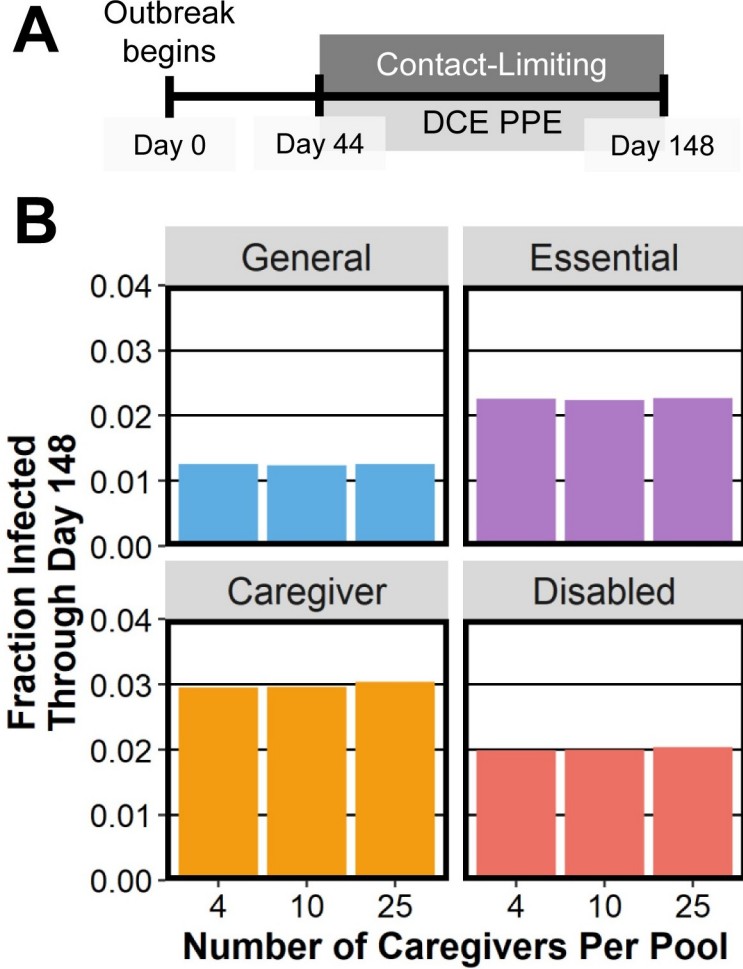

**Fig 8. The effect of the number of caregivers (4, 10, or 25) that are assigned to a given disabled person on the mean fraction of each subpopulation that becomes infected.** The label "DCE PPE" refers to the (D+C+E) mask-wearing scenario.

ratio of the upper bound to the baseline value is the same as the ratio of the baseline value to the lower bound. As expected, reducing the probability of breaking weak contacts when ill (see Fig 9A) and reducing mask effectiveness (see Fig 9B) both increase the number of infections. Importantly, however, varying these parameters does not affect the overall pattern of infections; in particular, caregivers remain the most vulnerable subpopulation. The total number of caregivers that became infected does not change when we increase the number of caregivers, but the fraction of caregivers that become infected decreases (see Fig 9C). A relatively large increase in the fraction of the population that serve as caregivers (from 0.0210 to 0.0281, which is a roughly 33% increase) leads to a relatively small decline in the number of disabled people who become infected (from 1,494 to 1,325, which is a roughly 11% decrease). The general and essential-worker subpopulations also experience fewer infections when we increase the fraction of caregivers in the population. The order of the risk levels of the different subpopulations remains the same for all scenarios. Overall, we find that our general conclusions are not affected by moderately varying these three uncertain parameters.

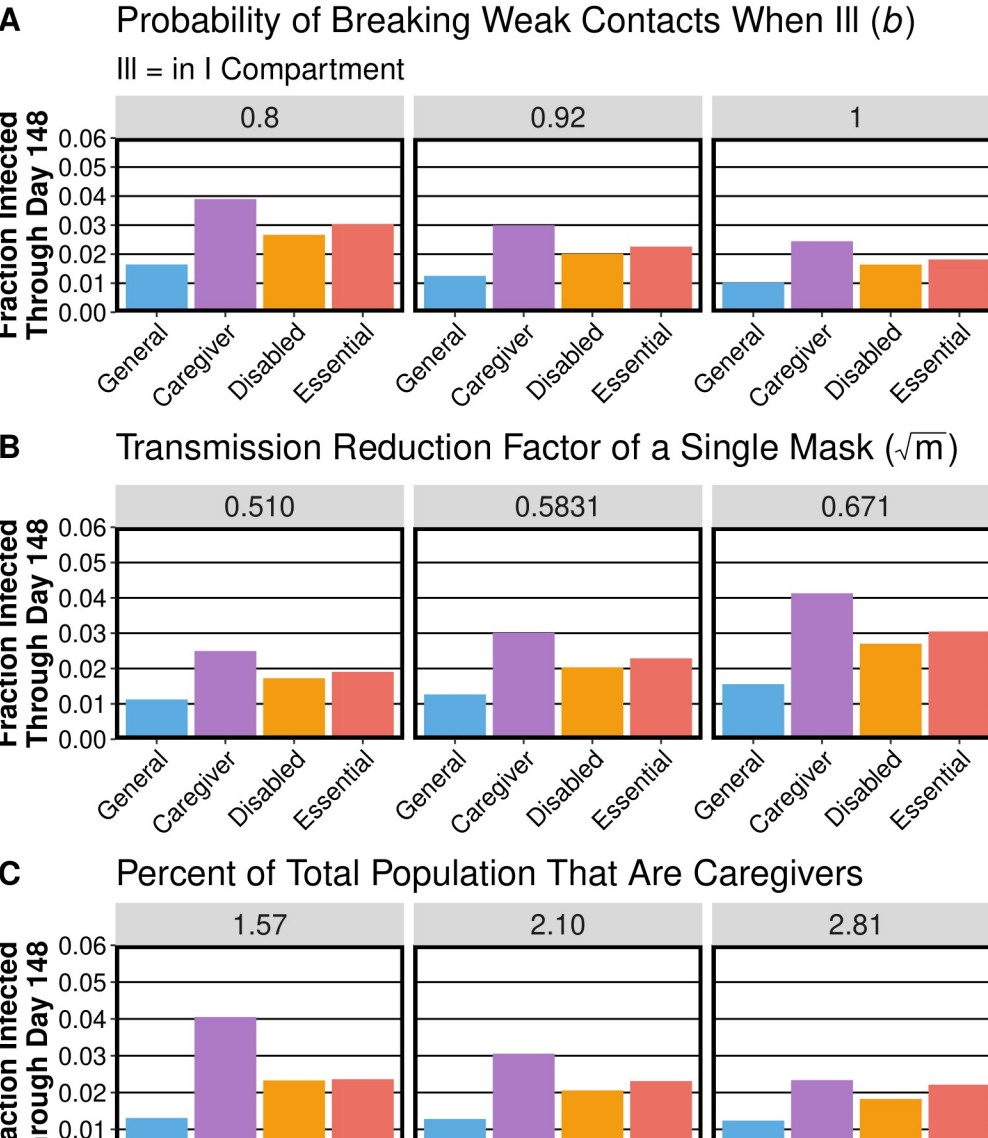

**Fig 9. The effects of (A) the probability of breaking weak contacts when ill, (B) mask effectiveness on the mean fraction that each subpopulation becomes infected, and (C) the percent of the population that serve as caregivers.**

Having observed that caregivers are the most likely of the four examined subpopulations to become infected with COVID-19 across all tested parameter sets, we investigate whether or not caregivers are also the most prone to spreading COVID-19. To do this, we seed all initial infections in a single subpopulation, rather than distributing the initially infected individuals uniformly at random across our model city's entire population. We calculate the mean fraction of each subpopulation that is infected cumulatively through 148 days. We find that the caregivers are the most potent spreaders of COVID-19, with each subpopulation reaching its highest infection rate when only caregivers are infected initially (see Fig 10). Seeding all initial

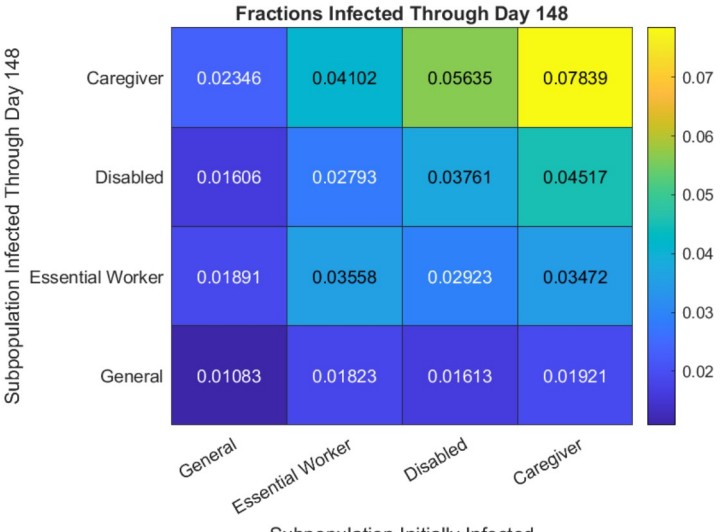

**Fig 10. The fraction of each subpopulation that is infected through day 148 when all of the initially infected individuals are in a single subpopulation.** On day 44, all groups limit contacts and the (D+C+E) mask-wearing scenario begins.

infections only in the disabled subpopulation leads to the second-largest number of infections in the caregiver subpopulation.

As we explain in Section A of the S1 Text, because of the intimacy of interactions between caregivers and disabled people, the relative risk of such an interaction is likely higher than is the case for typical household interactions. In Section C of the S1 Text, we also consider $w_c = 1$ (i.e., the risk level of a caregiver–disabled interaction is the same as that of a household interaction) and $w_c = 1.5$ (i.e., the risk level of a caregiver–disabled interaction is only moderately higher than that of a household interaction). When $w_c = 1$, essential workers are the most potent disease spreaders to all subpopulations except for the spread of the disease from caregivers to other caregivers. However, when $w_c = 1.5$, caregivers are the most potent disease spreaders to the disabled subpopulation and to themselves, and essential workers are the most potent disease spreaders to the general population and to themselves. This suggests that our conclusions about the impact of the caregiver subpopulation on the disabled subpopulation are plausible even if the relative risk $w_c$ is only moderately larger than 1.

Our finding that caregivers are the subpopulation that is most prone to spreading COVID-19 has potential implications for vaccine prioritization because vaccinating caregivers can indirectly protect other subpopulations. Because initial vaccine supplies are often extremely limited, we test the efficacy of vaccinating only a small fraction of the total population. To do this, we simulate the distribution of a very limited number of vaccines—an amount that is equivalent to enough vaccines for half (i.e., 10,151) of the mean remaining susceptible caregivers on day 148 (this is equal to approximately 1% of the total city population)—by moving a uniformly random subset of either susceptible caregivers, susceptible disabled people, susceptible essential workers, or susceptible members of the general population immediately to the removed compartment. When there are fewer susceptible people than people to move in a subpopulation, we move everyone in that subpopulation (and no other individuals) to the removed state. We also simulate a scenario with no vaccination. We simulate reopening at the same time as vaccination. In a reopening, all subpopulations return to their original weak-contact distributions, but all people wear masks during all non-household interactions. (For a timeline, see Fig 11A.) We

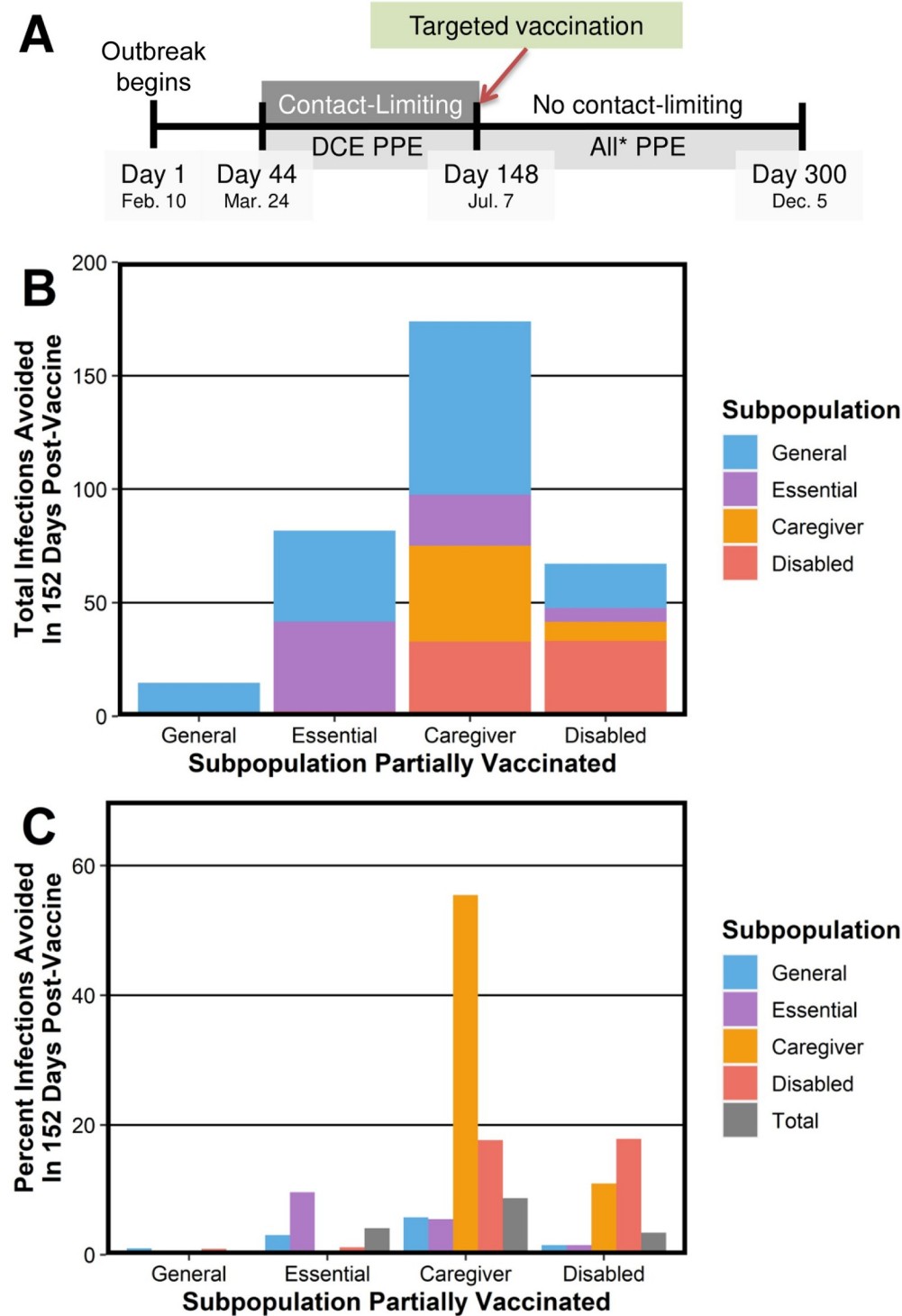

**Fig 11. The infections that are prevented in each subpopulation when one subpopulation is vaccinated with a limited number of vaccines.** (A) Timeline of contact-limiting and reopening in our simulations. After targeted vaccination occurs on day 148, there are no contact-limiting measures, but all individuals wear PPE during non-household interactions. (B) The total number of infections that are avoided between day 148 and day 300 in each subpopulation after vaccinating a limited number of individuals in a given subpopulation. (C) The percent of infections that are avoided in each subpopulation between day 148 and day 300 after vaccinating a limited number of individuals in a given subpopulation.

simulate our stochastic model of infections until day 300 and calculate the number of infections that are potentially preventable through the above vaccination strategies by comparing the results of these simulations to simulations that do not incorporate vaccination. This enables us to evaluate the benefits that vaccinating each subpopulation confers indirectly to other subpopulations.

Consistent with our previous findings, vaccinating caregivers prevents the largest number of infections. In our simulated scenario, targeting limited vaccinations to the caregiver subpopulation leads to a drop in total infections of 8.7% in comparison to the scenario without vaccination (see Fig 11B and 11C). It is second-most effective to vaccinate essential workers (this prevents 4.1% of the total infections) and third-most effective to vaccinate the disabled subpopulation (which prevents 3.4% of the total infections). Vaccinating the same number of individuals in the general population prevents only 0.7% of the total infections.

Vaccinating caregivers is an effective strategy to protect disabled people. When 10,151 caregivers are vaccinated, we reduce infections in disabled people by a mean of 17.7%. Vaccinating the same number of disabled people reduces infections by 17.9% (i.e., almost an equal number) in the disabled subpopulation. These almost equal effect sizes may arise from the relative sizes of the caregiver and disabled subpopulations in our model. Vaccinating 10,151 individuals entails vaccinating exactly half of the remaining susceptible caregivers, but 10,151 individuals constitutes only about 14% of the disabled subpopulation. Therefore, when the number of vaccines is extremely limited, vaccinating caregivers may be comparably effective at protecting the disabled population as directly vaccinating disabled people.

Notably, vaccinating either the caregiver subpopulation or the disabled subpopulation is much more effective at protecting the disabled subpopulation than vaccinating the essential-worker subpopulation, which prevents only 1.1% of the infections in the disabled subpopulation. Vaccinating caregivers even spares slightly more members of the general population than vaccinating essential workers; about 5.8% of the general-population infections are prevented when 10,151 caregivers are vaccinated, whereas about 3.4% of general-population infections are prevented when 10,151 essential workers are vaccinated. In our case study, the essential-worker subpopulation is the only subpopulation for whom the best strategy (of those that we considered) is to vaccinate the essential-worker subpopulation. With this strategy, vaccination prevents 9.6% of essential-worker infections, which is better than the 5.5% that are prevented when the caregiver subpopulation is vaccinated (see Fig 11C).

In our case study, we find that vaccinating the disabled subpopulation does not protect the caregiver subpopulation as effectively as vaccinating caregivers protects the disabled subpopulation. When 10,151 disabled people are vaccinated, a mean of about 11.0% of the caregiver cases are prevented. When the same number of caregivers are vaccinated instead, about 55.5% of the caregiver cases are prevented (see Fig 11). This fivefold difference may arise from the relative sizes of the caregiver and disabled subpopulations. Because a relatively small fraction of the disabled people with whom any given caregiver interacts are vaccinated and caregivers are often in the pools of multiple disabled people, our case study suggests that caregivers' risks are mitigated only slightly when only a small fraction of the disabled subpopulation are vaccinated.

## 4 Discussion

We now summarize and discuss our key results.

### 4.1 Our most significant findings

**Caregiver and disabled populations are extremely vulnerable to COVID-19 infections.** We simulated the spread of COVID-19 on networks to evaluate how vulnerable four

interconnected subpopulations—caregivers, disabled people, essential workers, and the general population—are to infection. Across multiple simulation conditions, we found that caregivers have the highest risk of infection and that disabled people have the second-highest risk of infection. This observation arises from multiple structural factors in our contact networks. First, there are many fewer caregivers than disabled people, so each caregiver typically has contact with multiple disabled people. This is reflected by caregivers having the largest number of direct neighbors and neighbors of neighbors. Second, caregiver–disabled contacts are stronger than other contacts, which (along with the large number of direct contacts of caregivers) contributes to caregivers also having the edges with the largest mean weights. Third, some of our simulations involved a contact-limiting phase, in which individuals reduce their number of weak contacts; however, caregiver–disabled contacts do not break during this phase. These structural factors render caregivers and disabled people particularly vulnerable to infection with COVID-19. We also found that caregivers are the most potent spreaders of COVID-19 once they are infected, and we suggest that this is due to the same factors (specifically, being well-connected in a social network) that make them most vulnerable to becoming infected. This agrees with the observations of Gozzi et al. [84], who examined two different spread-limiting strategies in an activity-driven network model and found that the most active nodes that do not comply with a spread-limiting strategy are the major drivers of disease spread. Reassuringly, our findings are robust to changes in the parameters—the effectiveness of masks, the probability of breaking contacts when ill, and the fraction of the population who are caregivers—in which we had the most uncertainty.

In our model, we assumed that the transition rate from the ill compartment to the hospitalized compartment is the same for all subpopulations. We also did not model death. Disabled people are more likely than other individuals to experience accessibility barriers to receiving healthcare and to have medical conditions that predispose them to severe cases of COVID-19 [85]. Additionally, caregivers are more likely than other individuals to belong to marginalized groups that are at increased risk due to systemic structural barriers to accessing medical care. Taking these factors into account may reveal an even more disproportionate disease burden on caregivers and disabled people. Ortega Anderez et al. [86] observed that small decreases in the exposure of medically vulnerable subpopulations significantly decreases overall mortality, underscoring how critical it is to identify interventions that effectively protect caregivers and disabled people.

**Effective interventions.** It is essential that the necessary medical services that at-home caregivers provide to disabled people continue to be available during a pandemic. These services are essential for survival; going without caregiving services endangers a disabled person's health. Therefore, we tested the effectiveness of various NPIs at preventing the spread of COVID-19 in these subpopulations. We found that mask-wearing during contacts between caregivers and disabled people is a very effective strategy for reducing infections in both subpopulations. This finding agrees with recent agent-based [87, 88] and bond-percolation [89] models of mask-wearing interventions. We recommend that home-healthcare agencies provide their employees with masks and (whenever possible) mandate their use on the job.

Additionally, we found that contact-limiting by disabled people alone only slightly reduces their risk of contracting COVID-19 if it is not accompanied by contact-limiting in the rest of a population. When all subpopulations limit contacts, cases of infection in the disabled and caregiver subpopulations fall by almost 50%. This result underlines the fact that changes in behavior in the general population can drive changes in disease spread in the disabled subpopulation. Disabled people alone are not numerous enough to change large-scale epidemic dynamics with their behaviors, and they are vulnerable to increases in disease spread that can occur when the general population changes its behavior. In the context both of the current

COVID-19 pandemic and possible future pandemics, we emphasize the critical influence of behavior by the general population on disabled communities. Mitigation efforts by the general population, such as contact-limiting (as in the present study), can protect disabled people much more than interventions in only the disabled subpopulation.

**Vaccinating caregivers shields other subpopulations, including disabled people.** A major application of modeling the spread of a disease on a network is evaluating strategies for targeted vaccination [55, 56, 90]. Prior research suggests that, under certain conditions, the largest eigenvector centrality of a network helps determine a network's threshold (e.g., in the form of a basic reproduction number) for a widespread outbreak of a disease [83]. This suggests that vaccinating nodes with large eigenvector centralities may be a useful control strategy. Several COVID-19 vaccines have been approved for use [91–94], and we sought to determine the most effective vaccination strategy in the context of our model. As a first step, we calculated the eigenvector centralities of the nodes in the network's four subpopulations. We calculated that essential workers have the largest mean eigenvector centrality in a single simulated population and that caregivers have the largest modal eigenvector centrality in the same simulated population. This result is a direct consequence of the contact distributions of these two subpopulations. For example, essential workers are sometimes in very large workplaces and sometimes in very small workplaces, whereas caregivers almost always work with multiple disabled people.

Investigating network structure alone in our model did not resolve which subpopulation is the most efficient one to vaccinate. Therefore, we analyzed how the dynamics of disease spread were affected by selectively vaccinating a subset of each of the subpopulations. We considered a hypothetical vaccine that is completely effective and permanently prevents any individual who receives it from contracting or spreading the virus SARS-CoV-2. Although this is unrealistic—vaccinated people can still contract SARS-CoV-2 and even spread it to others [95]—vaccinated people are much less likely than unvaccinated people to be diagnosed with the disease COVID-19 [96]. They also experience a faster drop in viral load when they are infected, so transmission periods may be shorter in vaccinated people [97]. Vaccine effectiveness against household transmission that leads to COVID-19 infection in vaccinated individuals was estimated at 71% in one study [98]. However, this study was conducted when the Alpha variant (Pango lineage designation B.1.1.7) of SARS-CoV-2 was predominant, and it is unknown whether this finding holds for the Delta variant (Pango lineage designation B.1.617.2) or other variants. Because new variants emerge frequently and vaccine adherence, availability, and manufacturers vary worldwide, we chose to examine a simplistic scenario instead of attempting to model any specific real-world situation.

We measured the effectiveness of vaccination strategies by comparing the numbers of infections in scenarios with and without vaccination. The number of infections that are avoided includes both infections that are prevented directly (specifically, when an individual who would have become infected had already received a vaccine) and ones that are prevented indirectly (specifically, some chains of transmission do not occur because individuals who would have spread the virus were instead vaccinated against it). Our simulations suggest that vaccinating caregivers (1) prevents the largest total number of infections and (2) prevents the most infections in three of the four subpopulations. (The exception is the subpopulation of essential workers.) In our simulations, vaccinating a specified number of caregivers protected an almost equal number of disabled people from infection (because of indirect prevention) as vaccinating the same number of disabled people.

It is necessary to be cautious when interpreting our findings about the relative efficiency of vaccinating different subpopulations. To obtain our results, we assumed that vaccines prevent the spread of COVID-19 from a vaccinated individual to other individuals. In a scenario in

which vaccines prevent serious illness but have no effect on viral transmission from vaccinated individuals, it is likely better to employ them in populations (e.g., disabled people) that are more likely to experience hospitalization and death. Moreover, even if vaccinating caregivers does turn out to be the most efficient way to reduce total case numbers of COVID-19, it may still be more ethical to prioritize vaccinating individual disabled people, particularly those who are elderly or have conditions that predispose them to severe disease [86]. In the real world, vaccination campaigns must balance many factors—including medical risk, public health, and equity—when assigning priority [99]. Additionally, we reiterate that the precise conclusions about vaccination strategies from our model may not hold in real-world scenarios. For example, it is important to consider a variety of local factors, including the amount of vaccine that is available, the relative sizes of the caregiver and disabled populations, and the distributions of ages and pre-existing conditions in these populations.

When a small number of caregivers serve a large number of disabled people who are not at particularly high medical risk, vaccinating caregivers has several benefits: (1) it directly protects caregivers, who often are in demographic groups with an elevated risk of COVID-19 complications; (2) it indirectly shields the disabled people for whom they care; and (3) it prevents the disruption of essential caregiving services to disabled people when caregivers are infected and must quarantine. Furthermore, for disabled people who cannot gain the benefit of vaccination—whether due to access issues with transportation or at vaccination centers, immunosuppression, or other health challenges—our findings suggest that it may be useful to provide caregivers with priority access to vaccines.

Our model strongly suggests that caregivers of disabled people are at increased occupational risk of both contracting and spreading COVID-19 and that protecting caregivers also provides substantial, quantifiable benefits to the vulnerable population that they serve. Therefore, we suggest that it should be a high priority for caregivers to be among the groups with early access to vaccines.

Especially when vaccines are not readily available, we emphasize the importance of continuing effective NPIs, such as mask-wearing and contact-limiting, in all subpopulations (including the general population). Additionally, vaccination campaigns should make it a priority to protect disabled people, and they should consider early vaccination of caregivers and disabled people as one potential strategy among continued society-wide NPIs to accomplish this goal.

### 4.2 Limitations and future directions

In interpreting our results, we made many assumptions to construct a tractable model to study. Accordingly, our results occur in the context of a variety of hypotheses about the epidemiology of COVID-19 in the disabled community and optimal strategies to mitigate the spread of the disease. Although we consider our hypotheses to be reasonable ones, we obviously did not perfectly describe the complexity of COVID-19, how it spreads, and how human behavior affects its spread. (See [100] for a recent review and agenda for integrating social and behavioral factors into models of disease spread.) We encourage readers to look at our paper's referee reports, which are publicly available, to examine referee comments about our paper's limitations, including those in the final publication.

In reflecting on our assumptions and our modeling (of both network structure and the spread of COVID-19), there are a variety of natural steps to take to enhance our work (beyond using disease-spread models with more compartments and reinfection). Although they are beyond the scope of the present paper, we elaborate on some of them. We encourage careful examination of the following ideas:

- Incorporating skilled nursing facilities and hospitals: We assumed that caregivers provide at-home care to disabled people. There are many disabled people who live in skilled care facilities, which have different care-giving and care-receiving networks than the ones that we examined.

- Lack of entry into and exit from a city: We did not consider the possibility that people enter our model city and introduce infections into its population. We also did not consider infected people who leave the city. This type of effect was studied in [49].

- Uncertainty in the numbers of disabled people and caregivers in a population: There is a lot of uncertainty in the proportions of disabled people and caregivers in a population. Unfortunately, there is not much reliable information about how many disabled people receive assistance for their activities of daily living and how many people in society serve as caregivers (possibly in an unpaid or undocumented capacity). It is very important to obtain more data about this and to incorporate it into modeling efforts.

- More precise distributions of weak contacts: It was very difficult for us to estimate the contact distributions of people before and during a lockdown, and it was even more difficult to estimate the level of contact-limiting. It is worthwhile to study the effects of different types of distributions of weak contacts. We briefly explore this issue in Section C of the S1 Text.

- Incorporating daily randomness of interactions: During each phase of our model COVID-19 pandemic, we fixed the set of potential daily contacts (they are only potential contacts because illness can temporarily sever ties) of our population's individuals, except for interactions between disabled people and caregivers. (We assigned a random caregiver from a pool to each disabled person.)

- Modeling contact changes during a city's reopening: One limitation of our network model is that when we assigned additional contacts to individuals after our model city reopens, we did so in a random way (for simplicity), rather than having individuals resume the contacts that they had before a lockdown. This choice mixes the contacts in the network, and it seems important to study the consequences of this choice.

- Heterogeneity in mask effectiveness: We assumed that all masks give the same transmission-reduction benefits. However, this is not realistic. There are a large variety of mask types and some people do not wear masks correctly, so it seems worthwhile to examine how heterogeneity in mask effectiveness affects disease dynamics.

- Modeling mask-compliance probabilistically: For a given type of interaction, we assumed that all individuals of a given subpopulation either wear masks or don't wear masks. In reality, only some fraction of a subpopulation will wear masks.

- Studying the importance of caregivers to disease spread: We speculated that the large modal eigenvector centrality of caregivers causes them to be more potent than other subpopulations at spreading COVID-19 infections. It seems useful to further investigate the importance of caregivers to disease spread.

- Temporal variations in infectivity during the course of an infection: We assumed that an infected individual has the same level of infectivity throughout their entire infectious period. We recognize that this is not the case.

- Modeling vaccination outcomes: We assumed simplistically that vaccination fully prevents COVID-19 infection and transmission. In reality, vaccination provides robust but incomplete protection from COVID-19. Vaccinated individuals can experience asymptomatic or

symptomatic disease and can transmit the virus to others, although at lower rates than unvaccinated individuals [12]. Our model does not account for infection of or transmission by vaccinated individuals.

- Effects of new variants of SARS-CoV-2: The SARS-CoV-2 virus has mutated with time, and some of our parameter estimates surely depend on specific strains of the virus and differ across both time and geographic regions.

- Uncertainties in timing: We used the simplistic assumption that all positive tests of COVID-19 of individuals in the I and H compartments occur at the beginning of an individual's first day in the relevant compartment. We also assumed that the availability of COVID-19 tests was the same throughout the first 148 days of the COVID-19 pandemic. Neither of these assumptions is realistic, and it seems worthwhile to consider more realistic testing scenarios.

## 4.3 Conclusions

We constructed a stochastic compartmental model of the spread of COVID-19 on networks that model a city of approximately 1 million residents and used it to study the spread of the disease in disabled and caregiver communities. Our model suggests that (1) caregivers and disabled people may be the most vulnerable subpopulations to exposure in a society (at least of the four subpopulations that we considered); (2) mask-wearing appears to be extremely effective at reducing the numbers of infections in caregivers and disabled people; (3) contact-limiting by an entire population appears to be far better at protecting disabled people than contact-limiting only by disabled people; and (4) caregivers may be the most potent spreaders of COVID-19 infections, and giving them and disabled people who need caregivers priority access to vaccines can help protect disabled people.

## Supporting information

**S1 Text. This file contains the parameter estimation and specific steps that we use in our simulations.**
(PDF)

## Acknowledgments

We gratefully acknowledge Deanna Needell and Sherilyn Tamagawa for making the introductions that allowed our team to form, and we thank Stephen Campbell (Data and Policy Analyst at PHI) for directing us to useful resources and helping refine our questions.

## Author Contributions

**Conceptualization:** Thomas E. Valles, Hannah Shoenhard, Joseph Zinski, Sarah Trick, Mason A. Porter, Michael R. Lindstrom.

**Data curation:** Thomas E. Valles, Michael R. Lindstrom.

**Formal analysis:** Thomas E. Valles, Michael R. Lindstrom.

**Funding acquisition:** Mason A. Porter.

**Investigation:** Thomas E. Valles, Michael R. Lindstrom.

**Methodology:** Thomas E. Valles, Michael R. Lindstrom.

**Project administration:** Michael R. Lindstrom.

**Resources:** Mason A. Porter.

**Software:** Thomas E. Valles, Hannah Shoenhard, Joseph Zinski, Michael R. Lindstrom.

**Supervision:** Mason A. Porter, Michael R. Lindstrom.

**Validation:** Thomas E. Valles, Joseph Zinski.

**Visualization:** Thomas E. Valles, Hannah Shoenhard, Joseph Zinski.

**Writing – original draft:** Thomas E. Valles, Hannah Shoenhard, Joseph Zinski, Sarah Trick, Mason A. Porter, Michael R. Lindstrom.

**Writing – review & editing:** Thomas E. Valles, Hannah Shoenhard, Joseph Zinski, Sarah Trick, Mason A. Porter, Michael R. Lindstrom.

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
