## [Decision Letter · Decision Letter 0]

9 Apr 2021

Dear Dr Lindstrom,

Thank you very much for submitting your manuscript "Networks of Necessity: Simulating Strategies for COVID-19 Mitigation among Disabled People and Their Caregivers" for consideration at PLOS Computational Biology.

As with all papers reviewed by the journal, your manuscript was reviewed by members of the editorial board and by several independent reviewers. In light of the reviews (below this email), we would like to invite the resubmission of a significantly-revised version that takes into account the reviewers' comments.

We cannot make any decision about publication until we have seen the revised manuscript and your response to the reviewers' comments. Your revised manuscript is also likely to be sent to reviewers for further evaluation.

Sincerely,

Jacopo Grilli

Associate Editor

PLOS Computational Biology

Nina Fefferman

Deputy Editor

PLOS Computational Biology

Reviewer's Responses to Questions

**Comments to the Authors:**

Reviewer #1: The authors here present an agent-based model which is tailored for studying the impact of COVID-19 on different social groups living in Ottawa, Canadá. In particular, they divide the population into four different types of individuals: people with disabilities, caregivers, essential workers and the rest of the population. Likewise, different types of interactions, namely weak and strong ties, are introduced in order to capture how close the contacts are among these groups. Once the compartmental scheme of the model is defined and the impact of non-pharmaceutical interventions is incorporated on the dynamics, the model is validated by comparing 100 stochastic trajectories with the official cases reported by local authorities. After validating the model, the authors characterize the relevance of the different groups by studying different network centrality measures. Finally, they explore different scenarios to assess the impact of different measures such as increasing the pool of caregivers available for each disabled individual or deploying selective vaccination campaigns.

I have certainly enjoyed reading the manuscript. Despite the huge complexity of the model as a result of the large parameters' space, the authors managed to fix most of them according to data extracted or inferred from existing literature. In this sense, I would like to congratulate the authors for the thoroughness and transparency of the rationale behind the choice of each parameter which clearly facilitates the reading of the manuscript and understanding of the model. Moreover, the inclusion of the different types of agents constitute a novel contribution which allows for reflecting the nature of the interactions occurring among different social groups and the realistic constraints faced by non-pharmaceutical interventions. As shown by the authors, these constraints, such as the impossibility of weakening the close interactions among disabled people and their caregivers, can be crucial to understand the outcome of containment policies.

In my opinion, the manuscript is a valuable contribution for PLOS Computational Biology due to the relevance of the results and the theoretical advances introduced in the modeling of COVID-19 spread across specific setups. Nonetheless, I have some comments that I would like the authors to address prior to the publication of the manuscript.

- I think that the epidemiological times involved in the transitions from A to I compartments and from I to R are too long. Regarding the first one, the authors assume that once they individuals are infected, they are also infectious and spend a given time before developing symptoms, which is adapted from the median of the distribution of incubation periods reported in [64]. Actually, the infectious period before being symptomatic is always shorter than the incubation period since the viral load is not high enough at the early stages to allow them to transmit the virus. Furthermore, although from a clinical point of view, it takes 14 days to recover from COVID-19, the infectiousness of symptomatic individuals sharply decreases around 5 days after showing symptoms, being the latter the relevant time for the model. To compensate the long infectious periods, it seems the calibrated transmissibility (~0.01-0.02) is lower than the one used/calibrated in other studies like [Pullano et al. Nature 590 (2020)] or ref. [38] (~0.07-0.08). I would like the authors to comment the differences of the parameters with the ones reported in the literature and also the possible influence of the longer epidemiological times on the epidemic trajectories generated by the model. I guess that including a model with shorter infectious periods would allow the system to respond more sharply to containment policies, thus better capturing the shape of the data.

- The variance of the results presented by the authors when performing the stochastic simulations is remarkable. Does this variance comes from the power-law nature of the distribution of weak contacts across the population?

- The assessment of the vaccination strategies involving different groups is really interesting and clearly reveal the relevance of central agents/node (caregivers) in the network. However, they mostly rely on assuming that vaccination completely removes agents from the system which allows for break down the transmission chains connected by caregivers. While there is a global consensus on the efficacy of the vaccine in preventing severe/mild infections and despite some promising preliminary results, it is not totally clear whether COVID-19 vaccines significantly reduce the infectiousness of vaccinated individuals. I think that the authors should include a discussion on this topic as a possible limitation of the results of the manuscript.

As minor details:

- I would include in Fig. 4b a new panel comparing the new daily positive test according to both data and the model. In addition, for the sake of readability, I would use the date in the x-axis rather than the total number of days.

- It would be very illustrative to indicate in Fig. 2 who correspond with the weak and the strong caregiver respectively.

- I would rewrite of the paragraph indicating the higher susceptibility of Black and Hispanic people to COVID-19 to stress that it is related to the social circumstances usually affecting them rather than to their ethnicity per se; otherwise it can be misinterpreted.

- The authors should revise the references to check if the preprints have been already published. For example, ref. 38 should be updated, since the manuscript is already published in [Physical Review X 10 (4), 041055 (2020)].

Reviewer #2: The work proposes a model for the spreading of COVID-19 among disabled people and their caregivers. The authors implement a detailed agent based model with four different types of agents (general population, essential workers, caregivers and disabled people) also including the effect of non-pharmaceutical interventions like the use of masks and social distancing. The main findings are that the caregivers-disabled community is extremely vulnerable to COVID-19 and vaccination strategies aimed directly at caregivers would be highly effective in protecting both disabled people and the general population (even more than vaccinating disabled people directly).

The topic could be of some interest and the work focuses on a category (home caregivers) overlooked in the thousands of works published this period on COVID-19 epidemiology. Having said that, however, there are some points that make me doubt about the relevance and possible impact of the results.

Here there are my major and minor remarks:

- From the plethora of studies published in the last year about COVID-19 diffusion it is clear that the extremely variable latency period and prodromic infections play a crucial role in the spreading (see for example Ferretti et al. "The timing of COVID-19 transmission" medrxiv) delaying identification and reducing the effectiveness of interventions. Thus, is quite strange that a model so complicated in the representation of interaction networks employs a relatively unrealistic epidemic dynamics. Although I do not believe that the essence of the results would change, my suggestion is to move to a more realistic epidemic model (e.g. the one in Di Domenico et al. BMC Med 18, 240 2020) that includes, at least, exposed and prodromic infections.

- Along with the epidemic dynamics, I am also puzzled by some of the results presented in the manuscript:

- First of all, the large variability in the model runs when comparing with the real data (Fig.4). Simulations show extremely large confidence intervals even during the calibration phase (till day 97). Intervals should be quite small during calibration and become larger after that. This is quite strange especially considering that the authors are fitting four parameters. The causes could be, among many others: a model not able to fit the data (see my comment above), a poor choice of the parameters for the fitting or the fact that parameters are changing over time.

- Along the same line, I have some doubts about the choice of fitting the parameters over a period that covers both the early phases of the outbreak (without restrictions) and government interventions that drastically change evolution of the disease (i.e. the inflection point in the data around day 80). Some of the parameters could change between the two phases making the fitting more complicated. Thus, my suggestion is to try to fit only the first part of the data or, at least, limit the fit to day 60 or 65.

- Another point is about the number of first and second neighbors (and strength) in Fig. 5. I think this is crucial to assess the relevance of the work as the very large average degree of caregivers can explain all the results of the work: it is well known in the computational epidemiology literature that hubs in scale networks are more vulnerable than other nodes and several immunization strategies are based on identifying and vaccinate hubs first. What puzzles me is the huge difference in degree (more than double) between caregivers and the general population or essential workers and, even after reading three times section A of the SI, it is not clear to me if the difference is really present in the data (i.e. in reality caregivers have way more contacts than the rest of the population) or it is an artifact of the assumptions taken to build the network. My opinion is that the authors should try to clarify this point by testing different ways of building the networks or analyzing some type of demographic data. I understand that such analyses could be difficult and data hard (or even impossible) to find but, as I said, I find really strange that caregivers in real life have on average more than the double of contacts that essential workers or any other person. Moreover, something similar also happens for disabled people.

- The manuscript is well written and the presentation is, in general, clear. However, I have to admit it took me three readings of sec. 2 and the SI to understand the (quite complicated) model and still I am not sure I got all the details. I acknowledge that the authors tried to give enough details in the main text (e.g. in section 2.2) but my suggestion is to try to clarify more the network creation process that, right now, is quite confusing.

Reviewer #3: Simulating the spreading of COVID-19 in different scenarios has become a major necessity to understand the dependencies between the different intervening factors, and to be able to give informed advice to the health authorities. In this paper, the authors focus on the role of disabled people and their caregivers as spreaders and targets of COVID-19. This is important since disabled people cannot live without the support of their caregivers, and usually they are considered as highly vulnerable to the disease. Unfortunately, caregivers can also be the vectors of the disease, thus compromising their attendees. This work analyzes this kind of interrelation at the level of a city, with people that can be: disabled, caregivers, essential worker, and general population. Data from the city of Ottawa (Canada) has been used to establish the setup and calibrate the model: structure of the population, evolution of the pandemic, and non-pharmaceutical interventions (NPI). The result is a series of simulations which cover a large set of scenarios, which evaluate the impact of the different interventions and possible people behaviors on the selected collectives. Finally, the paper tries to find the best vaccination strategy.

The paper is very well-written and organized, the figures and tables are clear to convey the messages and results, and the methodology is in accordance with the standards for this kind of scientific works. However, since the simulations are trying to reproduce a real situation, and all the results depend on a large number of parameters and assumptions, we must be sure that all the important details cannot be disputed.

The epidemic model, though minimalist, is enough for the object of study and, to the best of my knowledge, all the selection, decision, and calculation of the parameters of the model are acceptable, in accordance with the current literature. Of course, there exists an important uncertainty regarding some of the parameters, but in general the selections could be considered as correct. Anyway, I have serious doubts on one of the parameters, which potentially could lead to important differences in the results, and on the quality of the calibration.

Let me start with the main issues:

1) I do not understand why you use weights in Eq. (1) instead of rates or probabilities. With weights, beta loses its traditional meaning of infection rate per contact, becoming a combination between the infection rate and a normalization constant. However, if beta were an infection rate, you could recover its value from the literature instead of relying in fitting it.

2) My main concern is with the weight for the caregivers. It has been set to 1.7, which means 70% higher than for household contacts. You just point to reference [62] but, looking at that reference, it is not easy to infer what have you done to assign this value. Moreover, I simply do not agree in assigning a larger weight to caregivers than to household contacts. I believe there is no possible reason for that. Given that COVID-19 is airborne (the evidence for this fact is overwhelming), and disabled people are going to have the contacts with household and caregivers’ counterparts usually in scarcely ventilated indoor places, the risk of getting infected is going to be equivalent in both cases. Household contacts should be the upper bound for infectious risk, since people at home do not wear masks, share the air for long times, and no social distance is maintained. Maybe there is a problem with the interpretation of the data in [62]. A larger secondary attack rate for caregivers should come from a larger number of contacts, not a higher risk of contagion, which is what should appear in Eq. (1). The only remaining factor to consider, the duration of the contact, should also be larger for household contacts than for caregivers.

3) In view of my previous concern, I would suggest to also check the value assigned to weak contacts.

4) I would say that most of the obtained results heavily depend on this assignment of such a large weight for caregivers, thus the conclusions could require big adjustments.

5) The deviation between data and simulations (Fig. 4) is quite large. If instead of plotting the accumulated cases (not recommended) you plot the new daily cases, the deviations would probably be more evident. In general, for a good calibration of the model, you need good data. It is well-known that, in the initial phase of the pandemic of COVID-19, the capacity of testing was very low in most countries, and only people with symptoms were detected; remember that for some time it was even argued that asymptomatic COVID-19 was not possible, and if so, some believed they could not be contagious. This means that the real number of cases could have been much larger than the detected through the scarce number of tests, thus compromising the calibration of the model. I do not know if this happened in Ottawa, but at least it must be taken into consideration. I would have expected a large underestimation of new cases at the beginning, but good agreement at the end. This is not what we observe in Fig. 4.

6) A better calibration approach could have been the use of the new daily hospitalizations or new daily deaths instead of new cases, since those data are much more reliable. Unfortunately, the model merges fatalities with recoveries, thus making impossible to use deaths for its calibration and/or validation, and probably daily hospitalizations would require to separate ICUs from non-ICUs, depending on the data availability.

The summary of the previous issues could be that any conclusion on COVID-19 requires a good calibration and fitting of the parameters of the model, and I have serious doubts on them.

Other minor issues:

1) The introduction is now a bit old: currently, there are several vaccines available for COVID-19. An update of the whole text to the current situation would be a good idea, even if the paper is limited to the analysis of the first wave.

2) It would be helpful to establish the correspondence between dates (10 February, 24 March, etc.) with the number of days that appear in the plots (1, 44, etc.).

3) Section 2.2: “essential workers either do or do include mask-wearing” -> “essential workers either do or do NOT include mask-wearing”

4) Please specify the NPIs used in Fig. 4. Is it D+C+E after day 44?

5) When describing the structure of the network before and after the lockdown, it is not mentioned the specific days selected. I guess the structure is statistically equivalent for all the days in the same interval.

6) All the main issues described above arouse after looking at Fig. 5, in which the strength of caregivers and disabled are too different with respect to general and essential, impossible to understand and believe.

7) The criteria that disabled population has the same mean value for weak contacts as the general population seems an overestimation: it seems more realistic that disabled people has less contacts.

8) In Fig. 12A, a dashed line on top of a solid line (with different widths) would allow to see both curves.

9) Please check the references, some of the preprints could have already been published.

**Have all data underlying the figures and results presented in the manuscript been provided?**

Reviewer #1: None

Reviewer #2: Yes

PLOS authors have the option to publish the peer review history of their article (what does this mean?). If published, this will include your full peer review and any attached files.

Reviewer #1: No

Reviewer #2: No

Reviewer #3: No

**Have the authors made all data and (if applicable) computational code underlying the findings in their manuscript fully available?**

Reviewer #3: Yes
---

## [Decision Letter · Decision Letter 1]

20 Dec 2021

Dear Dr Lindstrom,

Thank you very much for submitting your manuscript "Networks of Necessity: Simulating COVID-19 Mitigation Strategies for Disabled People and Their Caregivers" for consideration at PLOS Computational Biology. As with all papers reviewed by the journal, your manuscript was reviewed by members of the editorial board and by several independent reviewers. The reviewers appreciated the attention to an important topic.

All the reviewers are overall happy to the current version of the manuscript and they do nor raise any major issue which prevents publication. In addition to a few typos and formatting suggestions, there are only a few minor comments, mainly from reviewer 2, for which the author might want to opportunity to further clarify their modeling choices. We expect to accept this paper, even if you respond at this point saying you don't want to make any further changes, but wanted to give you the option of addressing these last points.

Please prepare and submit your revised manuscript (or response that you are happy with the paper as it is) within 30 days. If you anticipate any delay, please let us know the expected resubmission date by replying to this email.

Sincerely,

Jacopo Grilli

Associate Editor

PLOS Computational Biology

Nina Fefferman

Deputy Editor

PLOS Computational Biology

[LINK]

All the reviewers are overall happy to the current version of the manuscript and they do nor raise any major issue which prevents publication. In addition to a few typos and formatting suggestions, there are only a few minor comments, mainly from reviewer 2, for which the author might want to opportunity to further clarify their modeling choices.

Reviewer's Responses to Questions

**Comments to the Authors:**

Reviewer #1: The authors have addressed correctly my concerns and I therefore recommend this article for publication in PLOS Computational Biology.

Reviewer #2: Let me start by saying that I really appreciated the efforts made by the authors in trying to address my and the other reviewers comments. The work is now clearer and surely more solid. The manuscript went through a profound revision and now the many assumptions behind the model and the network are clearly stated. I also appreciate the choice of the authors to move to a more realistic epidemic dynamics that would help grasping the timescale of COVID-19 spreading.

Said that however, I think that some of the main issues raised by me and the other reviewers in the previous round have not been fully addressed (and, more importantly, I am not sure it would be possible to address them).

Summarizing, most of the results presented rely heavily on three main points:

- The epidemiological model that, in order to be able to make reliable predictions, should reproduce the dynamics of the disease.

- The fraction of individuals and the number of their contacts for the four categories considered.

- The weights assigned to each type of contact.

The problem is that all these points depend on assumptions/estimations that are hard to test or too sensible to changes/errors in the data.

Being more specific:

- The fitting of the model is still too poor to assure that it is able to reproduce CODIV-19 evolution. I understand the difficulties and I think the authors did their best to solve the issue however, given the huge differences between the curves in fig.4, I do not think that the model is capable of reproducing the dynamics of the disease (I am referring not only to the still quite large variability, but also to the differences in the final number of infected, etc.).

Along with many others, the cause could be in the reliability of the data. In this sense, I strongly support the suggestion of Reviewer 3 of using hospitalization and deaths data instead of case counts (a solution widely adopted in the literature) as, even if they could bring some biases (age differences and so on), they are way more solid than infection counts.

Another problem could be in the parameters they are trying to fit and not in their number. I am not sure that the probability of being tested ill and the maximum number of contacts are the best choices. A common practice in the literature is to employ a metapopulation model (so a well-mixed population at the lowest scale) and fit only one parameter, representing the transmission probability. A possible way of adapting this line of reasoning to the contact network proposed by the authors could be to fit the baseline infection probability and one of the weights used to model the contacts but, this is just a suggestion.

- The network building algorithm depends heavily on the estimation of the number of strong and weak contacts that, although based on Canadian census data, could be unreliable. For example, starting from data in refs 19 and 20 of the SI, the authors estimate that each disabled individual would see, on average, 2 caregivers. The authors then calculate O_c as the fraction of disabled people divided by the fraction of caregivers and multiply by 2, getting O_c = 6.95. Even if this value is way more reasonable than the one in the previous version of the work (due to a plotting error), it is still 1.7 contacts per day larger than the regular population and I still cannot find a reason for that. More importantly, given the small fractions of caregivers and disabled people, small errors in their number (the number of caregivers could be easily underestimated due to illegal work, etc.) could lead to quite large changes in O_c. This limitation has been correctly highlighted by the authors in the conclusions, however, I do not think that stating it is enough. A sensitivity analysis for most of these estimates should be done to check the robustness of the results.

- A similar line of reasoning holds for the weights of the different interactions (as pointed out by Reviewer 3). In their response to the comments raised by the Reviewer, the authors provide a compelling argument for assigning larger weights to caregivers interactions and I appreciate the sensitivity study conducted in this sense. However, there is little support to this assumption in the data or in the literature. Again, I trust the narrative proposed by the authors, but I think that more evidence is needed to justify it and small differences in these estimates could lead to larger errors in the results.

Finally, all these sources of uncertainty combined make me doubt the solidity of the results.

Minor remarks:

- In eq.1 \\sigma should be \\sigma_i

- On page 12 when discussing Fig. 5 the text states: "We find that caregivers have the most first-degree and second-degree contacts (see Fig. 5A,B)" while from the figure it seems that essential workers have the larger number of contacts. Probability this is an error from the previous version of the manuscript.

Concluding, although the authors made a huge work in spelling out all the assumptions and limitations of their model, my overall impression is that it relies too heavily on them to support solid conclusions.

Reviewer #3: The authors have made an extraordinary work dealing with my comments and those of the other reviewers, carefully addressing even the most secondary issues, and with a great scientific rigor. For sure, I can grant publication as is, and I apologize for the delay I could have added to the review process.

Minor comment: It would be useful that the Table and Figures in the Supporting Information were indexed as S1, S2, etc., to avoid confusion with the Table and Figures in the main text.

**Have the authors made all data and (if applicable) computational code underlying the findings in their manuscript fully available?**

Reviewer #1: Yes

Reviewer #2: Yes

Reviewer #3: Yes

PLOS authors have the option to publish the peer review history of their article (what does this mean?). If published, this will include your full peer review and any attached files.

Reviewer #1: No

Reviewer #2: No

Reviewer #3: No

Figure Files:

Data Requirements:

Reproducibility:

References:

---

## [Editor Report · Decision Letter 2]

21 Mar 2022

Dear Dr Lindstrom,

We are pleased to inform you that your manuscript 'Networks of Necessity: Simulating COVID-19 Mitigation Strategies for Disabled People and Their Caregivers' has been provisionally accepted for publication in PLOS Computational Biology.

Best regards,

Jacopo Grilli

Associate Editor

PLOS Computational Biology

Nina Fefferman

Deputy Editor

PLOS Computational Biology

---

## [Editor Report · Acceptance letter]

12 May 2022

PCOMPBIOL-D-20-02331R2 

Networks of Necessity: Simulating COVID-19 Mitigation Strategies for Disabled People and Their Caregivers

Dear Dr Lindstrom,

I am pleased to inform you that your manuscript has been formally accepted for publication in PLOS Computational Biology. Your manuscript is now with our production department and you will be notified of the publication date in due course.

With kind regards,

Livia Horvath
